# Neuronal activity determines distinct gliotransmitter release from a single astrocyte

Ana Covelo, Alfonso Araque*

Department of Neuroscience, University of Minnesota, Minneapolis, United States

**Abstract** Accumulating evidence indicates that astrocytes are actively involved in brain function by regulating synaptic activity and plasticity. Different gliotransmitters, such as glutamate, ATP, GABA or D-serine, released form astrocytes have been shown to induce different forms of synaptic regulation. However, whether a single astrocyte may release different gliotransmitters is unknown. Here we show that mouse hippocampal astrocytes activated by endogenous (neuron-released endocannabinoids or GABA) or exogenous (single astrocyte $Ca^{2+}$ uncaging) stimuli modulate putative single CA3-CA1 hippocampal synapses. The astrocyte-mediated synaptic modulation was biphasic and consisted of an initial glutamate-mediated potentiation followed by a purinergic-mediated depression of neurotransmitter release. The temporal dynamic properties of this biphasic synaptic regulation depended on the firing frequency and duration of the neuronal activity that stimulated astrocytes. Present results indicate that single astrocytes can decode neuronal activity and, in response, release distinct gliotransmitters to differentially regulate neurotransmission at putative single synapses.

DOI: https://doi.org/10.7554/eLife.32237.001

## Introduction

Astrocytes play key roles in brain homeostasis, such as, neurotransmitter uptake, water and ion balance, and blood flow control (*Simard and Nedergaard, 2004*; *Huang et al., 2004*). In addition, astrocytes are emerging as cells with relevant roles in synaptic function, being active players of the Tripartite Synapse (*Araque et al., 1999*). They express a wide variety of functional receptors that can be activated by neurotransmitters (*Araque et al., 2001*; *Haydon and Carmignoto, 2006*; *Perea et al., 2009*; *Volterra et al., 2014*; *Araque et al., 2014*; *Navarrete and Araque, 2010*; *Panatier et al., 2011*; *Volterra et al., 2005*; *Di Castro et al., 2011*; *Perea et al., 2016*; *Mariotti et al., 2018*), and they can release different gliotransmitters, such as glutamate, GABA, ATP and D-serine, which activate neuronal receptors (*Araque et al., 2014*; *Navarrete and Araque, 2010*; *Panatier et al., 2011*; *Volterra et al., 2005*; *Di Castro et al., 2011*; *Perea et al., 2016*; *Perea and Araque, 2007*; *Parri et al., 2001*; *Bezzi et al., 2004*; *Martín et al., 2015*; *Min and Nevian, 2012*; *Halassa and Haydon, 2010*) and regulate synaptic transmission and plasticity (*Panatier et al., 2011*; *Perea and Araque, 2007*; *Martín et al., 2015*; *Min and Nevian, 2012*; *Halassa and Haydon, 2010*; *Covelo and Araque, 2016*; *Henneberger et al., 2010*; *Min et al., 2012*). Numerous studies have reported different types of short- and long-term synaptic regulatory phenomena induced by specific gliotransmitters released from astrocytes in several brain areas (*Araque et al., 2014*). Yet, whether a single astrocyte releases different gliotransmitters or whether different astrocyte subpopulations release distinct gliotransmitters remains unknown. Furthermore, the cellular signaling processes that regulate the release of different gliotransmitters need to be identified to better understand the synaptic consequences of astrocytic activation on network function.

*For correspondence:
araque@umn.edu

**Competing interests:** The authors declare that no competing interests exist.

Calcium elevations evoked by activation of type 1 cannabinoid receptors (CB1Rs) (*Navarrete and Araque, 2010*; *Gómez-Gonzalo et al., 2015*; *Navarrete and Araque, 2008*), group I metabotropic glutamate receptors (mGluRs) (*Perea and Araque, 2005*; *Fellin et al., 2004*), muscarinic acetylcholine receptors (mAChRs) (*Navarrete et al., 2012*) or GABA$_B$ receptors (GABA$_B$Rs) (*Perea et al., 2016*; *Kang et al., 1998*) in hippocampal astrocytes have been shown to stimulate the release of glutamate (*Navarrete and Araque, 2010*; *Perea et al., 2016*; *Gómez-Gonzalo et al., 2015*; *Navarrete and Araque, 2008*; *Navarrete et al., 2012*; *Kang et al., 1998*). Astrocytic glutamate has been shown to transiently enhance the probability of neurotransmitter release at CA3-CA1 synapses through activation of type 1 receptors (mGluR$_1$s) (*Navarrete and Araque, 2010*; *Perea et al., 2016*; *Gómez-Gonzalo et al., 2015*; *Navarrete et al., 2012*), and potentiate or depress inhibitory transmission through activation of kainate or group II/III metabotropic glutamate receptors, respectively (*Kang et al., 1998*; *Liu et al., 2004a, 2004b*).

In contrast, hippocampal astrocytes have also been shown to release ATP which, after being degraded to adenosine, controls basal hippocampal synaptic activity (*Panatier et al., 2011*) and tonically depresses neurotransmission (*Pascual et al., 2005*). Astrocytic purinergic signaling also mediates heterosynaptic depression (*Pascual et al., 2005*; *Boddum et al., 2016*; *Serrano et al., 2006*; *Zhang et al., 2003*; *Chen et al., 2013*), which has been proposed to involve successive events that include activation of GABAergic interneurons, release of GABA that activates GABA$_B$R in astrocytes, and astrocytic Ca$^{2+}$ elevations that trigger the release of ATP which, after being converted to adenosine by extracellular ectonucleotidases, activates neuronal A$_1$Rs that depress synaptic transmission (*Boddum et al., 2016*; *Serrano et al., 2006*).

Using hippocampal astrocytes and CA3-CA1 synapses as a paradigmatic tripartite synapse model, we sought to determine whether single astrocytes release both glutamate and ATP and determine the neuronal stimulating conditions that control the differential release of these gliotransmitters. We found that single astrocytes may release both glutamate and ATP, which evoke a biphasic regulation of the probability of neurotransmitter release at single hippocampal synapses. The amplitude and temporal characteristics of this regulation depended on the neuronal activity pattern that stimulated astrocytes. These results indicate that astrocytes decode the time and frequency of neuronal activity, responding with the release of different gliotransmitters that differentially regulate synaptic function.

## Results

### A single synapse can be modulated by different gliotransmitters

Astrocytes are known to induce synaptic potentiation or heterosynaptic depression of CA3-CA1 synapses through the release of glutamate or ATP, respectively (*Navarrete and Araque, 2010*; *Perea and Araque, 2007*; *Pascual et al., 2005*; *Boddum et al., 2016*; *Serrano et al., 2006*; *Zhang et al., 2003*; *Chen et al., 2013*). To study whether single synapses can be regulated by distinct gliotransmitters, we used the minimal stimulation approach, which is considered to allow monitoring single or very few synapses (*Navarrete and Araque, 2010*; *Perea and Araque, 2007*; *Isaac et al., 1996*; *Raastad, 1995*; *Allen and Stevens, 1994*), and tested whether they could undergo both synaptic regulatory phenomena (mediated by astroglial glutamate and ATP/adenosine). We recorded pairs of CA1 pyramidal neurons and monitored unitary excitatory postsynaptic currents (EPSCs) evoked by minimal stimulation of Schaffer collaterals (SC)(*Figure 1*), and applied the two stimulation paradigms that are known to evoke either astrocyte-mediated synaptic potentiation or heterosynaptic depression (*Figure 2A*). First, we depolarized one neuron (from −70 to 0 mV, 5 s) to induce eCB release (*Navarrete and Araque, 2010*; *Navarrete and Araque, 2008*; *Chevaleyre and Castillo, 2004*; *Kreitzer and Regehr, 2001*; *Wilson and Nicoll, 2001*; *Ohno-Shosaku et al., 2002*) and quantified parameters of synaptic transmission in the other neuron to assess the eCB-induced synaptic potentiation (*Navarrete and Araque, 2010*). Then, we applied a high-frequency stimulation (HFS; 3 trains of stimuli at 100 Hz for 1 s, delivered every 30 s) in an independent set of SC synapses to elicit the heterosynaptic depression (*Pascual et al., 2005*; *Serrano et al., 2006*) (*Figure 2A,B,H*). Consistent with previous reports (*Navarrete and Araque, 2010*), neuronal depolarization (ND) induced a transient increase in the probability of release (n = 5 neurons; p=0.018) and synaptic efficacy (p=0.029) without significant changes in the synaptic

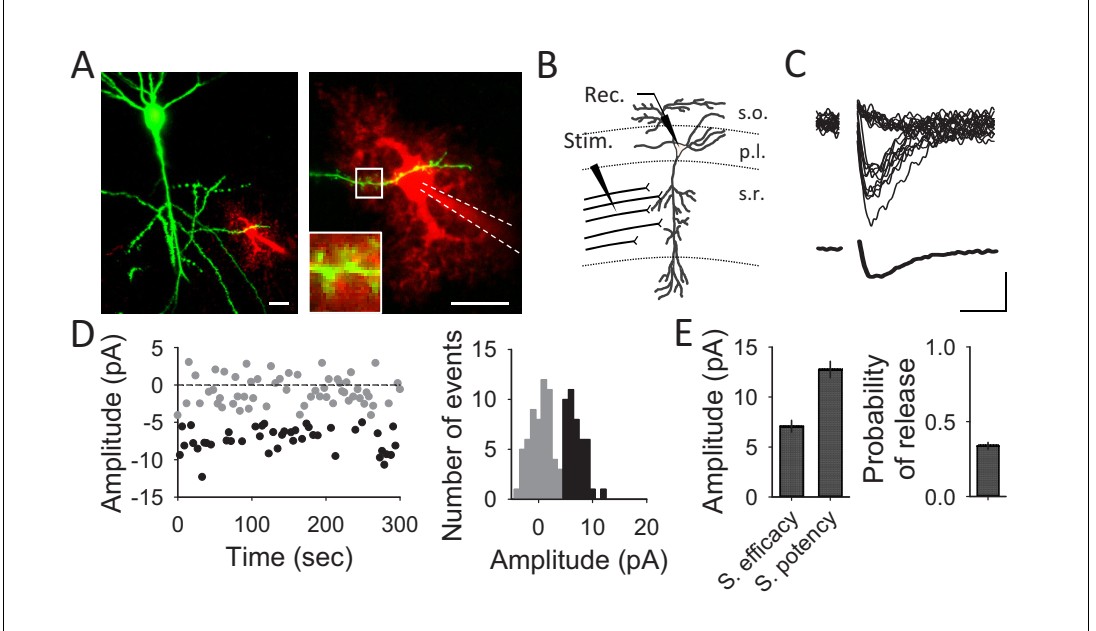

**Figure 1.** Unitary CA3-CA1 synapse recordings from hippocampal pyramidal neurons. (**A**) Fluorescent images showing a pyramidal cell expressing GFP (green) and an astrocyte loaded with Texas Red (red) through the recording pipette. Scale bars: 20 µm. (**B**) Schematic drawing depicting a recorded CA1 pyramidal neuron and the stimulating electrode. (**C**) Representative EPSC traces obtained with the minimal stimulation technique showing successes and failures in neurotransmitter release (n = 20; upper panel) and the average trace acquired from those traces (synaptic efficacy; lower panel). Scale bars: 5 pA, 10 ms. (**D**) EPSC amplitudes of a representative synapse that showed failures (grey) and successes (black) in neurotransmitter release (left panel) and its corresponding histogram (right). (**E**) Synaptic parameters (synaptic efficacy, synaptic potency and probability of release) obtained from representative recordings (n = 40 synapses).

DOI: https://doi.org/10.7554/eLife.32237.002

potency (p=0.187; *Figure 2C,D*). The HFS protocol induced a longer lasting transient decrease in the probability of release (p=0.039) and synaptic efficacy (p=0.013) without significant changes in the synaptic potency (p=0.321; *Figure 2C,D*). Both the ND-evoked synaptic potentiation and the HFS-evoked depression were observed in 31.25% (5 out of 16 neurons) of putative single synapses (*Figure 2F*). As expected, the ND-evoked synaptic potentiation was prevented in the presence of the mGluR$_1$ antagonist LY367385 (100 µM; n = 4 neurons; p=0.928; *Figure 2E–G*) and in the astroglial GFAP-CB1-null mice (*Han et al., 2012*; *Martin-Fernandez et al., 2017*) (n = 10; p=0.8) (*Figure 2—figure supplement 1A–D*), which specifically lack CB1Rs in GFAP-expressing astrocytes (cf. *Figure 2—figure supplement 2*). To confirm that astrocytes in GFAP-CB1-null mice selectively lacked CB1Rs, we tested the astrocyte responsiveness to the CB1R agonist WIN55-212,2 and to ATP in slices obtained from WT and GFAP-CB1-null mice. While astrocytes from GFAP-CB1-null mice did not respond to WIN55-212,2 (n = 91 astrocytes from n = 10 slices, p=0.462), they show robust calcium responses to ATP (p<0.001), indicating that astrocytes remained functional but selectively lack CB1R-mediated responses. Moreover, CB1R-mediated neuronal responses in these mice were also assessed by monitoring the depolarization-induced suppression of inhibition (DSI), a process known to be mediated by presynaptic activation of CB1 receptors (*Chevaleyre and Castillo, 2004*; *Kreitzer and Regehr, 2001*; *Wilson and Nicoll, 2001*; *Ohno-Shosaku et al., 2002*). Both wildtype and GFAP-CB1-null mice showed similar DSI, indicating that CB1Rs were specifically absent in astrocytes in GFAP-CB1-null mice (*Figure 2—figure supplement 2*). In contrast to ND-evoked synaptic potentiation, the HFS-mediated heterosynaptic depression was blocked by the A$_1$ receptor (A$_1$R) antagonist CPT (2 µM; n = 6 neurons; p=0.253) (*Figure 2E–G*) but still present in the GFAP-CB1-null mice (n = 5; p=0.004; *Figure 2—figure supplement 1E–H*). These results indicate that a putative

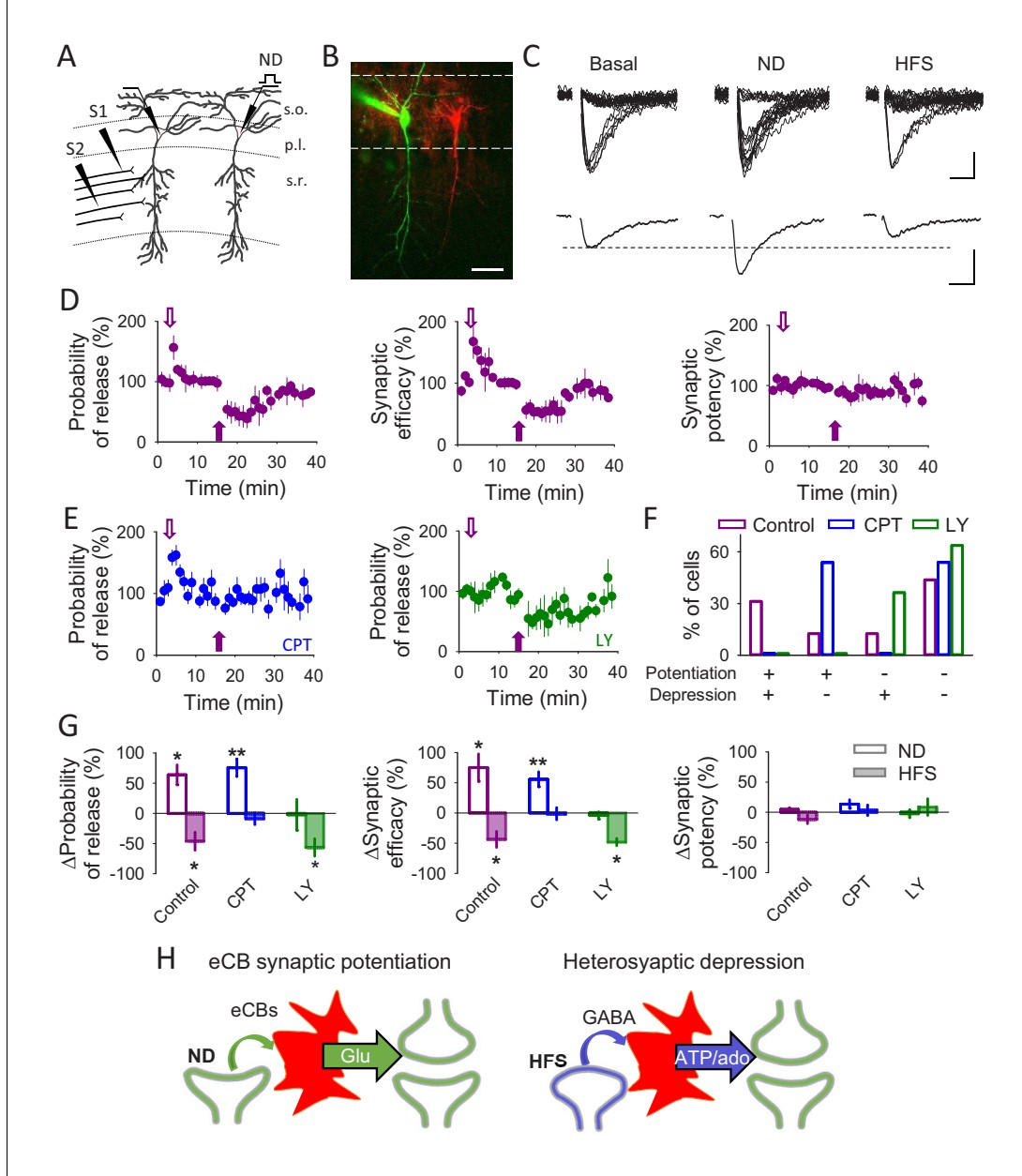

**Figure 2.** A single synapse can be modulated by different gliotransmitters. (**A**) Schematic drawing depicting paired recordings from CA1 pyramidal neurons and the stimulating electrodes (S1 and S2). (**B**) Fluorescent image showing two pyramidal cells, one loaded with Texas Red (red) and the other one with Alexa 488 (green). Scale bar: 40 µm. (**C**) Representative EPSC traces (n = 20) and their average traces during basal conditions, after a neuronal depolarization (ND) and after a HFS. Scale bars: 5 pA, 10 ms. (**D**) Synaptic parameters vs time in control conditions. Open and filled arrows indicate ND and HFS, respectively. (**E**) Probability of release vs time in the presence of CPT (2 µM) or LY367385 (100 µM). Open and filled arrows indicate ND and HFS, respectively. (**F**) Percentage of cells that underwent synaptic potentiation, depression, both or no change in the different conditions. (**G**) Relative changes in synaptic parameters after ND and HFS in control conditions (n = 5) and in the presence of CPT (2 µM; n = 6) or LY367385 (100 µM; n = 4). (**H**) Schematic summary depicting the signaling pathways leading to eCB-mediated synaptic potentiation or heterosynaptic depression. Data are represented as mean ± s.e.m., *p<0.05, **p<0.01.

DOI: https://doi.org/10.7554/eLife.32237.003

The following figure supplements are available for figure 2:

**Figure supplement 1.** eCBs mediate glutamate-induced synaptic potentiation but not heterosynaptic depression.
DOI: https://doi.org/10.7554/eLife.32237.004

**Figure supplement 2.** Astroglial specific GFAP-CB1-null mice shows impairments in astrocytic responsiveness to CB1 receptor agonists but not in the neuronal CB1 receptor dependent depolarization-induced suppression of inhibition (DSI).

*Figure 2 continued on next page*

*Figure 2 continued*

DOI: https://doi.org/10.7554/eLife.32237.005

single synapse can be modulated by both glutamate and ATP/adenosine released by astrocytes in response to ND or HFS, respectively.

While some studies proposed that heterosynaptic depression involves GABA$_B$R-mediated astrocytic release of ATP after interneuron activation by HFS of SC (*Figure 2H*) (*Serrano et al., 2006*), other studies have reported a GABA$_B$R-mediated astrocytic glutamate release by direct interneuron stimulation that leads to a synaptic potentiation (*Perea et al., 2016*; *Kang et al., 1998*). Because those reports used two different forms of interneuron stimulation, that is, direct interneuron depolarization or synaptic stimulation, we hypothesized that the differences in the type of gliotransmitter released (glutamate *vs* ATP/adenosine) depend on the interneuron firing pattern. To test the effects of interneuron activity on astrocytic responses and CA3-CA1 synaptic transmission, we recorded pairs of interneurons and CA1 pyramidal neurons. We monitored Ca$^{2+}$ signals in *stratum radiatum* astrocytes and the SC-induced unitary EPSCs in the pyramidal cell, and stimulated interneurons with a train of short depolarizing pulses to trigger action potentials at 20 Hz (for 90 s), as previously reported for interneurons (*Hu et al., 2014*; *Pike et al., 2000*) (*Figure 3A,B,G*). Interneuron stimulation induced astrocyte Ca$^{2+}$ elevations (n = 143 astrocytes in n = 17 slices; p<0.001; *Figure 3C–F*) that were prevented by the GABA$_B$R antagonist CGP54626 (1 µM; n = 85 astrocytes in n = 9 slices; p=0.795) but not by the GABA$_A$R antagonist picrotoxin (n = 157 astrocytes in n = 16 slices; p<0.001; *Figure 3F*), indicating that interneurons stimulate astrocytes through GABA$_B$R activation. In addition, interneuron activity induced a relatively fast transient potentiation of synaptic transmission that was followed by a longer-lasting synaptic depression (n = 5 neurons; Pr potentiation p=0.012; Pr depression p=0.024; *Figure 3H–J*). Both phenomena were unaffected by picrotoxin (n = 6 neurons; Pr potentiation p=0.002; Pr depression p<0.001) but abolished by CGP54626 (n = 9 neurons; Pr potentiation p=0.25; Pr depression p=0.849; *Figure 3J*), indicating that they were selectively mediated by activation of GABA$_B$Rs. To confirm GABA$_B$Rs involvement in the astrocyte Ca$^{2+}$ signals and the biphasic synaptic regulation we applied baclofen (90 s), a GABA$_B$R agonist, into the *stratum radiatum* while monitoring astrocyte Ca$^{2+}$ and synaptic currents in the pyramidal neuron (*Figure 3—figure supplement 1A*). Baclofen application induced Ca$^{2+}$ increases in 49 ± 8% of the recorded astrocytes (n = 61 astrocytes in n = 11 slices; p<0.001; *Figure 3—figure supplement 1B*) and a biphasic synaptic regulation (n = 6; Pr potentiation p<0.001; Pr depression p=0.034; *Figure 3—figure supplement 1C*), mimicking results obtained with interneuron depolarization. Taken together, these results suggest that astrocytes mediate interneuron-induced potentiation and depression of transmitter release. To test this idea, we used the inositol-1,4,5-triphosphate (IP$_3$) receptor type 2 knock-out (IP$_3$R2-null) mice (*Li et al., 2005*), in which G protein-mediated Ca$^{2+}$ mobilization is selectively impaired in astrocytes (*Di Castro et al., 2011*; *Martín et al., 2015*; *Gómez-Gonzalo et al., 2015*; *Navarrete et al., 2012*; *Petravicz et al., 2008*). In these mice, interneuron depolarization failed to both elevate astrocytic Ca$^{2+}$ (n = 74 astrocytes in n = 10 slices; p=0.719; *Figure 3E,F*) and induce the potentiation or depression of neurotransmitter release (n = 9 neurons; Pr potentiation p=0.749; Pr depression p=0.489; *Figure 3J*).

We further tested the astrocyte involvement on the synaptic biphasic regulation loading the astrocytic network with GDPβS through a patch pipette to specifically block G protein signaling in astrocytes. In GDPβS-loaded slices interneuron depolarization failed to induce astrocyte Ca$^{2+}$ elevations (n = 60 astrocytes in n = 8 slices; p=0.53; *Figure 3F*) and to induce the biphasic synaptic regulation (n = 7 neurons; Pr potentiation p=0.3; Pr depression p=0.538; *Figure 3J*). In addition, we used GABA$_B$-null mice that has been reported to lack GABA$_B$ receptors specifically in astrocytes (*Perea et al., 2016*). We first confirmed that GABA$_B$-mediated astrocyte responses were absent in these transgenic mice. While astrocytes from GABA$_B$-null mice did not respond to baclofen (n = 74 astrocytes from n = 6 slices, p=0.418), they responded to ATP (p<0.001), indicating that astrocytes were functional but selectively lacked GABA$_B$R-mediated responses. Furthermore, neurons from wildtype and GABA$_B$-null mice similarly responded to baclofen, indicating that GABA$_B$Rs were specifically absent in astrocytes in GABA$_B$-null mice (*Figure 3—figure supplement 2*). In this transgenic line, interneuron stimulation failed to induce Ca$^{2+}$ elevations in astrocytes (n = 53 astrocytes in

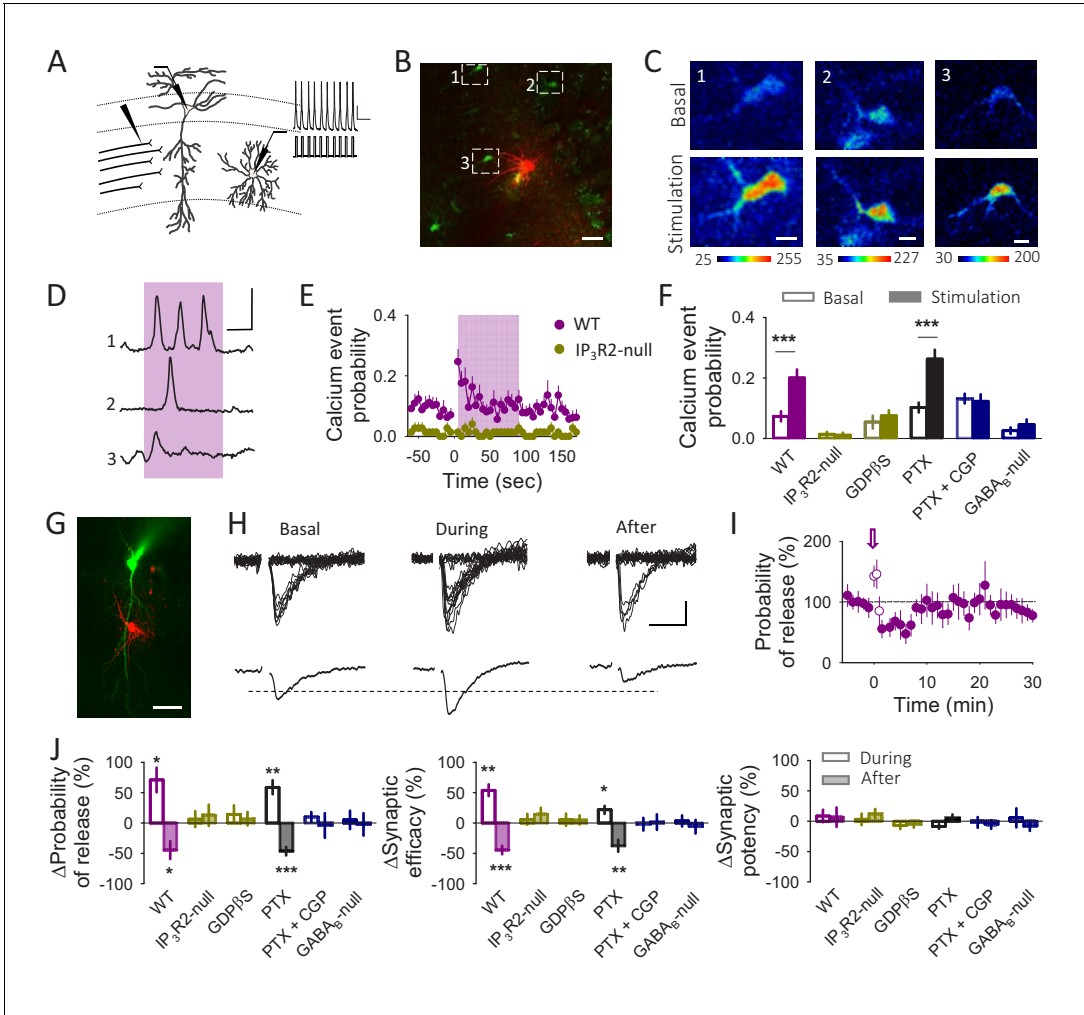

**Figure 3.** Astrocytes differentially modulate synaptic transmission in response to neuronal activity. (**A**) Schematic drawing depicting paired recordings from a CA1 pyramidal neuron and a *S. R.* interneuron and the stimulating electrode. Representative traces showing the interneuron depolarization train are displayed as insets. (**B**) Fluorescence image showing an interneuron loaded with Texas Red (Red) and astrocytes loaded with Fluo4-AM (Green). Scale bar: 40 μm. (**C**) Pseudocolor images showing the fluorescence intensities of Fluo4-AM of the 3 astrocytes marked in panel B before (basal) and during the interneuron stimulation. Scale bars: 10 μm. (**D**) Representative calcium traces obtained from the 3 astrocytes marked in panel B. The pink area represents the stimulus duration (interneuron depolarization, 20 Hz during 90 s). Scale bar: 40%, 30 s. (**E**) Calcium event probability vs time in the WT and in the IP$_3$R2-null mice. The pink area corresponds with the duration of the stimulus. (**F**) Changes in the calcium event probability 15 s before (Basal) and 15 s after the stimulation started (stimulation) in control conditions (n = 143 astrocytes), in the IP$_3$R2-null mice (n = 74 astrocytes), in slices loaded with GDPßS (10 mM; n = 60 astrocytes), in the presence of picrotoxin (PTX, 50 μM; n = 157 astrocytes) or CGP54626 (1 μM; n = 85 astrocytes) and in the astroglial GABA$_B$-null (n = 53 astrocytes). (**G**) Fluorescence image of a CA1 pyramidal cell loaded with Alexa 488 (Green) and an interneuron loaded with Texas Red (Red). Scale bar: 50 μm. (**H**) Representative EPSC traces (n = 20) and their average traces before (basal), during and after the interneuron depolarization train. Scale bars: 5 pA, 20 ms. (**I**) Synaptic parameters vs time in control conditions. Zero time corresponds with the beginning of the IN depolarization train (arrow). The open circles represent the synaptic parameters measured during the IN depolarization train. (**J**) Relative changes in synaptic parameters during and after the IN depolarization train in control conditions (n = 5), in the IP$_3$R2-null mice (n = 9), in slices loaded with GDPßS (n = 7), in the presence of Picrotoxin (PTX, 50 μM; n = 6), CGP54626 (1 μM; n = 9) and in the astroglial GABA$_B$-null (n = 9). Data are represented as mean ± s.e.m., *p<0.05, **p<0.01, ***p<0.001.

DOI: https://doi.org/10.7554/eLife.32237.006

The following figure supplements are available for figure 3:

**Figure supplement 1.** Baclofen application mimics interneuron depolarization.
DOI: https://doi.org/10.7554/eLife.32237.007

**Figure supplement 2.** Astroglial specific GLAST-GABA$_B$-null shows impairments in astrocytic responsiveness to GABA$_B$ receptor agonists but not in the neuronal responsiveness.
DOI: https://doi.org/10.7554/eLife.32237.008

**Figure supplement 3.** The biphasic synaptic regulation is not cell type specific.

*Figure 3 continued on next page*

*Figure 3 continued*

DOI: https://doi.org/10.7554/eLife.32237.009

n = 10 slices; p=0.23; *Figure 3F*) and the biphasic synaptic regulation (n = 9 neurons; Pr potentiation p=0.75; Pr depression p=0.9; *Figure 3J*). Taken together, these data indicate that GABA$_B$-mediated astrocyte Ca$^{2+}$ signaling is necessary to induce the interneuron-evoked potentiation and depression of synaptic transmission.

To further test these results in identified interneurons, we used the PV$^+$ transgenic mice that express the fluorescent protein tdTomato in parvalbumin-positive interneurons (*Figure 3—figure supplement 3A,B*). In agreement with previous results, stimulation of identified PV$^+$ interneurons with trains of depolarizing pulses to fire action potentials at 20 Hz elicited the biphasic synaptic regulation (n = 5 neurons; Pr potentiation p=0.01; Pr depression p=0.001; *Figure 3—figure supplement 3C,D*), which was abolished in the presence of the GABA$_B$R antagonist CGP (n = 5 neurons; Pr potentiation p=0.821; Pr depression p=0.94; *Figure 3—figure supplement 3D*). These results indicate that interneuron-evoked stimulation of astrocytes leads to the potentiation and depression of synaptic transmission.

## A single astrocyte may release both glutamate and ATP/adenosine

We then investigated what type of gliotransmitters mediate the synaptic potentiation and depression evoked by interneuron depolarization. The mGluR$_1$ antagonist LY367385 (100 μM) prevented the initial potentiation but did not affect the delayed depression (n = 4 neurons; Pr potentiation p=0.976; Pr depression p<0.001; *Figure 4A,B*), which, in contrast, was abolished by the A$_1$R antagonist CPT (2 μM; n = 7 neurons; Pr potentiation p<0.001; Pr depression p=0.131; *Figure 4A,B*). These results, which are consistent with the effects of ND and HFS, respectively (see *Figure 2*), indicate that the synaptic potentiation and depression induced by interneuron activity are mediated by activation of mGluR$_1$ and A$_1$Rs, respectively. Notably, the interneuron-evoked astrocyte Ca$^{2+}$ signal was unaffected by the receptor antagonists (n = 96 astrocytes in n = 11 slices p<0.001 and n = 109 astrocytes in n = 11 slices p<0.001 in the presence of LY367385 and CPT, respectively; *Figure 4C,D*), suggesting that glutamate and ATP/adenosine are released downstream of the GABA$_B$-mediated astrocyte Ca$^{2+}$ signal. Taken together, these results indicate that interneuron activity activates astrocyte gaba$_B$Rs that elevate Ca$^{2+}$ and stimulates the release of glutamate and ATP/adenosine, leading to the biphasic modulation of synaptic transmission, that is, an initial glutamate-induced synaptic potentiation followed by a purine-induced synaptic depression.

Present results indicate that a putative single synapse can be regulated by both astrocytic glutamate and ATP/adenosine. Rodent hippocampal astrocytes occupy separate anatomical domains with minimal overlap between astrocytic processes (*Bushong et al., 2002*; *Halassa et al., 2007*; *Wilhelmsson et al., 2006*), thus it is likely that a single synapse may spatially interact with only a single astrocyte (see *Figure 1A*), which suggests that a single astrocyte releases different gliotransmitters. However, recent evidence suggests that astrocytes are a heterogeneous population of cells with different functional properties (*Khakh and Sofroniew, 2015*; *Chai et al., 2017*), suggesting that different gliotransmitters could be released from different astrocytes. To test these hypotheses, we stimulated a single astrocyte and monitored a putative single synapse (*Figure 5A*), using the astrocyte-mediated hippocampal synaptic regulation as a functional assay to test whether a single astrocyte can release both glutamate and ATP/adenosine.

To stimulate a single astrocyte, we loaded one astrocyte with the Ca$^{2+}$ cage NP-EGTA (5 mM) and Fluo4 (to monitor astrocyte Ca$^{2+}$) through the recording pipette and photo-stimulated with UV light to elevate Ca$^{2+}$ in the recorded astrocyte, bypassing the GABA$_B$R-induced Ca$^{2+}$ signaling. To ensure the stimulation of only the NP-EGTA filled astrocyte, the recording pipette also contained GDPßS (10 mM) to fill the gap junction-connected astrocytic syncytium (*Figure 5B*) to prevent G protein-mediated intercellular astrocytic signaling (*Navarrete et al., 2012*). UV light stimulation elevated astrocyte Ca$^{2+}$ levels in the NP-EGTA-loaded astrocyte (*Figure 5C,D*) but not in the surrounding astrocytes in the field of view (*Figure 5D,E*); probably due to the lack or insufficient gap junction diffusion of NP-EGTA into neighboring astrocytes. Light-evoked Ca$^{2+}$ elevations in a single astrocyte elicited an initial potentiation followed by a longer-lasting depression of the synaptic

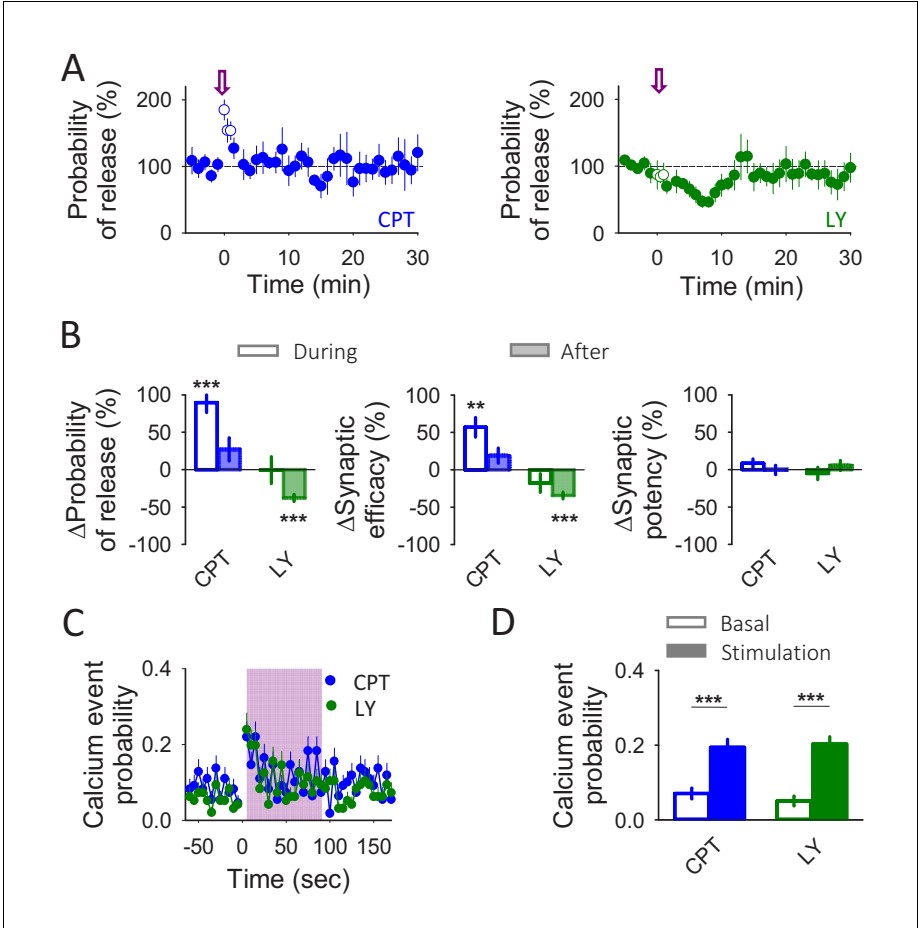

**Figure 4.** Astrocytes release glutamate and ATP/adenosine in response to neuronal activity. (**A**) Synaptic parameters vs time in the presence of CPT (2 μM) or LY367385 (100 μM). Zero time correspond with the beginning of the IN depolarization train (arrow). The open circles show the synaptic parameters measured during the IN depolarization train. (**B**) Relative changes in synaptic parameters during and after the IN depolarization train in the presence of CPT (2 μM; n = 7) or LY367385 (100 μM; n = 4). (**C**) Calcium event probability vs time in the presence of CPT (2 μM) or LY367385 (100 μM). The pink area corresponds with the duration of the stimulus. (**D**) Changes in the calcium event probability 15 s before (basal) and 15 s after the stimulation started in the presence of CPT (2 μM; n = 109 astrocytes) or LY367385 (100 μM; n = 96 astrocytes). Data are represented as mean ± s.e.m., *p<0.05, **p<0.01, ***p<0.001.
DOI: https://doi.org/10.7554/eLife.32237.010
The following figure supplement is available for figure 4:

**Figure supplement 1.** Astrocytes co-release glutamate and ATP.
DOI: https://doi.org/10.7554/eLife.32237.011

transmission (n = 9 neurons; Pr potentiation p<0.001; Pr depression p<0.001; *Figure 5F,G*), as similarly observed by synaptically activated astrocytes (see e.g. *Figure 3I*). In the absence of NP-EGTA, UV light stimulation did not affect the synaptic parameters (n = 7 neurons; Pr potentiation p=0.894; Pr depression p=0.551; *Figure 5G*; cf.[*Perea and Araque, 2007*]). In addition, light-evoked Ca$^{2+}$ uncaging rescued the biphasic synaptic modulation in the IP$_3$R2-null mice (n = 6 neurons; Pr potentiation p<0.003; Pr depression p=0.002; *Figure 5G*), indicating that astrocytic glutamate and ATP/adenosine are release downstream of the astrocyte Ca$^{2+}$ signaling. Therefore, activation of a single astrocyte is sufficient to induce glutamate- and purine-mediated synaptic modulation, indicating that one astrocyte can release at least two different gliotransmitters.

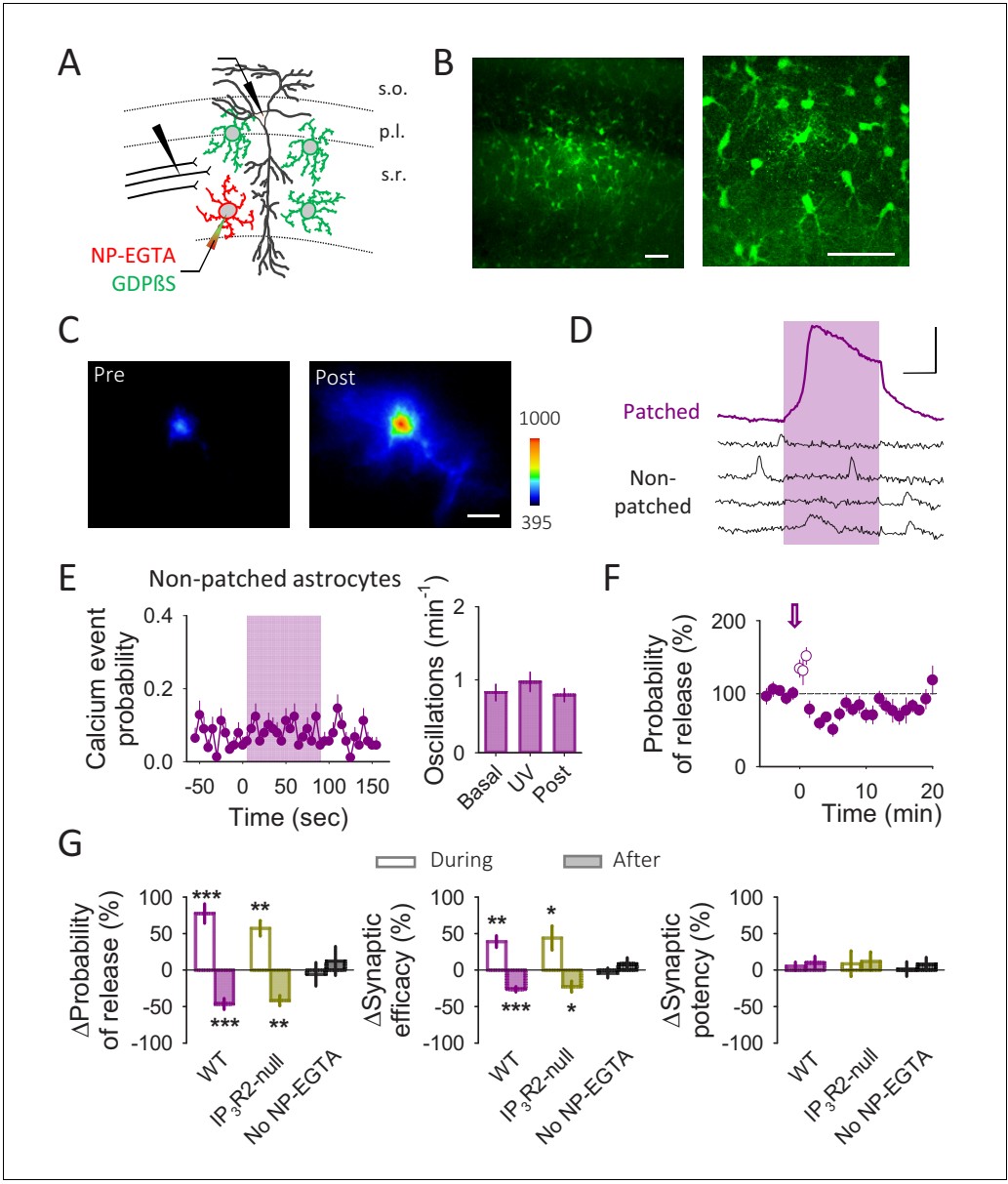

**Figure 5.** A single astrocyte regulates single synapses downstream of the GABAergic activity. (**A**) Schematic drawing depicting paired recordings from one pyramidal neuron and one astrocyte filled with NP-EGTA, GDPβS filling the astrocytic network, and the stimulating electrode. (**B**) Fluorescence images of an astrocyte network loaded with biocytin through the patched astrocyte. Scale bars: 50 μm. (**C**) Pseudocolor images representing fluorescence intensities of a Fluo4 loaded astrocyte before (pre) and after (post) UV light-induced calcium uncaging. Scale bar: 10 μm. (**D**) Representative calcium traces of NP-EGTA loaded astrocyte (patched; purple trace) and other astrocytes in the field of view loaded with GDPßS (non-patched; black traces). The pink area represents the UV light stimulation. Scale bar: 50%, 30 s. (**E**) Calcium event probability vs time (left panel) and number of calcium oscillations per minute (right panel) observed in the non-patched astrocytes. (**F**) Probability of release vs time. Zero time corresponds with the beginning of the calcium uncaging (arrow). The open circles show the probability of release measured during UV light illumination. (**G**) Relative changes in synaptic parameters during and after UV light stimulation of astrocytes filled with NP-EGTA (5 mM) and GDPßS (10 mM) recoded form WT (n = 9) and IP$_3$R2-null mice (n = 6) and in the absence of NP-EGTA (n = 7). Data are represented as mean ± s. e.m., *p<0.05, **p<0.01, ***p<0.001.
DOI: https://doi.org/10.7554/eLife.32237.012

## Time- and frequency-dependence of gliotransmitter release

Present results indicate that glutamate released from astrocytes had a relatively rapid and transient effect on synaptic transmission when compared to ATP/adenosine effects, which occurred with a delayed and slower time course. We then investigated whether the astrocytic synaptic regulation mediated by glutamate and ATP/adenosine was time- and frequency-dependent on neuronal activity. We stimulated interneurons with trains of 20 Hz depolarizing pulses of different duration (100 ms, 1 s, 30 s, 60 s and 90 s) and analyzed the synaptic parameters in the recorded pyramidal neuron (*Figure 6A*). A relatively short stimulation train (100 ms) was insufficient to affect the probability of neurotransmitter release (n = 6 neurons; Pr potentiation p=0.945; Pr depression p=0.87; *Figure 6B*), however, interneuron depolarization for 1 s induced the transient synaptic potentiation but not the synaptic depression (n = 6 neurons; Pr potentiation p<0.001; Pr depression p=0.852; *Figure 6B*). Notably, this synaptic potentiation lasted longer than the potentiation observed during the biphasic phenomenon (3.5 ± 0.56 min and 0.93 ± 0.2 min for 1 and 90 s stimulation, respectively; *Figure 6C*), suggesting that the astrocytic glutamate effect is curtailed by the ATP/adenosine effect. This idea is further supported by the fact that the duration of the potentiation was longer after being pharmacologically isolated than during the biphasic modulation when the delayed depression occurred (*Figure 4—figure supplement 1B–D*).

Moreover, purine-induced depression of neurotransmitter release required relatively long depolarizing trains (30, 60 and 90 s; *Figure 6B,C*). These results indicate that relatively short stimuli are

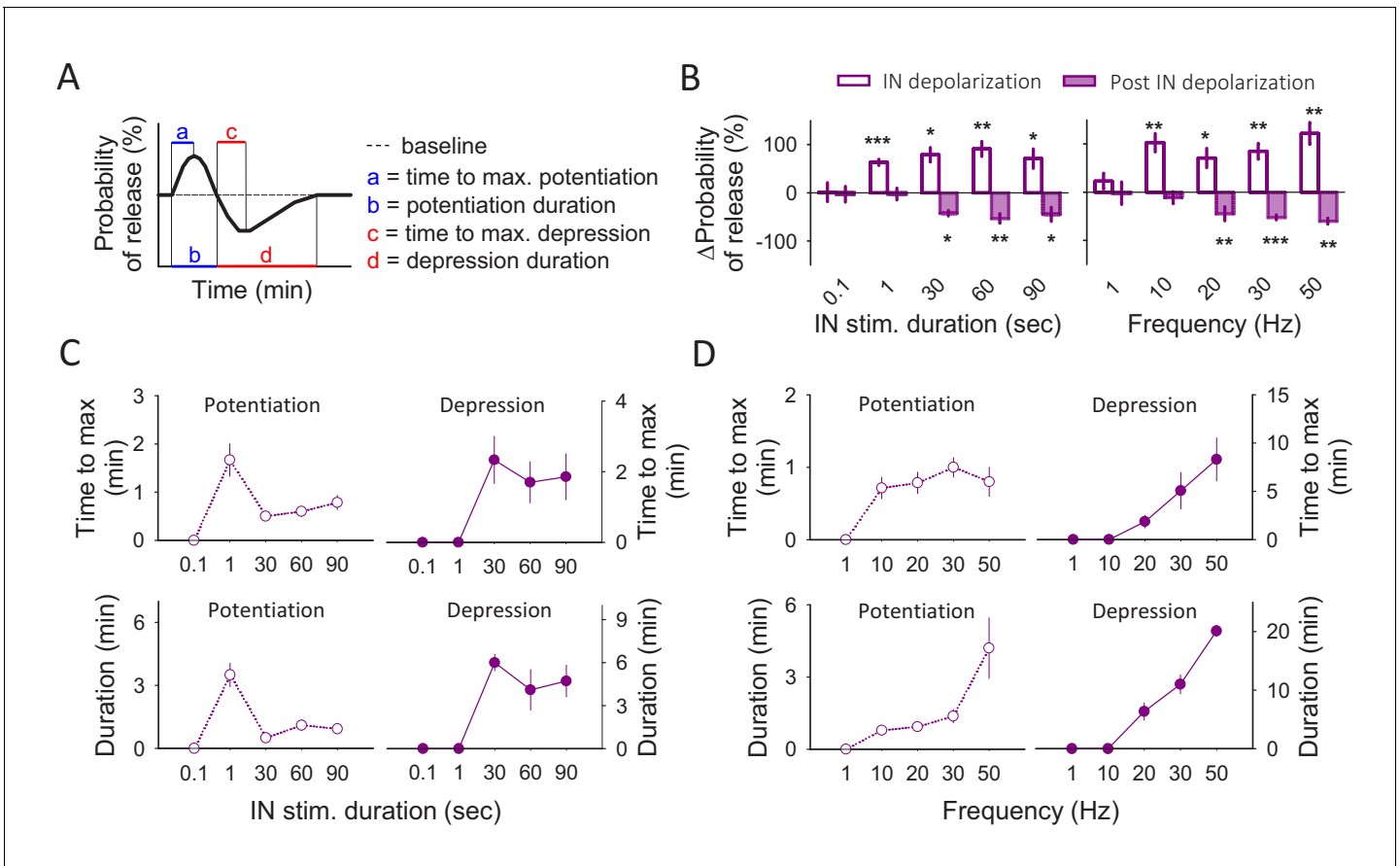

**Figure 6.** Frequency and stimulus duration dependence of gliotransmitter release. (A) Schematic illustration of the astrocyte-induced synaptic modulation and the parameters measured. (B) Percentage of effect (potentiation or depression) when the interneurons were depolarized at 20 Hz for different time windows (left panel) and when interneurons were depolarized during 90 s at different frequencies (right panel). (C) Time to maximum effect and effect duration with the different stimulation durations. (D) Time to maximum effect and effect duration with the different stimulation frequencies. Data are represented as mean ± s.e.m., *p<0.05, **p<0.01, ***p<0.001.
DOI: https://doi.org/10.7554/eLife.32237.013

sufficient to induce glutamate- but not ATP/adenosine-mediated synaptic regulation by astrocytes, whereas sustained neuronal activity is required to elicit astrocytic ATP/adenosine-mediated synaptic depression.

We finally investigated the dependence of the biphasic synaptic regulation on the interneuron firing frequency. We depolarized interneurons to fire action potentials at different frequencies (1, 10, 20, 30 or 50 Hz for 90 s). At a relatively low firing rate (1 Hz), neither the potentiation nor the depression of synaptic transmission were observed (n = 8 neurons; Pr potentiation p=0.722; Pr depression p=0.937; *Figure 6B*), indicating that a relatively low interneuron activity is insufficient to activate neuronal receptors by gliotransmitters. At 10 Hz firing rate, only the synaptic potentiation was present whereas the synaptic depression was absent (n = 7 neurons; Pr potentiation p=0.002; Pr depression p=0.407; *Figure 6B*), suggesting a lower threshold for the glutamate-mediated synaptic regulation than for ATP/adenosine-mediated synaptic depression. At 20 Hz, the synaptic potentiation reached a maximum at 0.79 ± 0.15 min and lasted for 0.93 ± 0.2 min, when the synaptic depression started. The synaptic depression lasted for 6.36 ± 1.47 min reaching a maximum at 1.86 ± 0.65 min. At 30 and 50 Hz both the potentiation and the synaptic depression lasted longer (synaptic potentiation duration 1.38 ± 0.28 min and 4.2 ± 1.26 min, respectively; synaptic depression duration 11 ± 1.6 min and 20.1 ± 0.9 min, respectively; *Figure 6B,D*), indicating that stronger neuronal activity leads to a longer effect of gliotransmission. In summary, for both the duration and firing rate of interneuron activity, the amplitude of the synaptic potentiation and depression is relatively constant beyond a threshold value, which is lower for the potentiation than for depression for both the duration and firing rate of interneuron activity (*Figure 6B*). Regarding the duration of interneuron activity dependence, once the synaptic depression appears, its time to maximum and its duration are relatively constant. In contrast, the time to maximum and the duration of the synaptic potentiation is higher at a relatively lower stimuli duration, indicating that it is curtailed by the synaptic depression (*Figure 6C*). Regarding the dependence on interneuron firing rate, the time to maximum of the synaptic potentiation is relatively similar at the different frequencies, but its duration increased as the firing rate increased. On the contrary, the time to maximum and the duration of the synaptic depression is increased as the firing rate increased (*Figure 6D*).

Taken together, these results indicate that the effects of gliotransmission on synaptic transmission were time- and activity-dependent on the neuronal firing, suggesting that astrocytes decode both the temporal and frequency characteristics of the neuronal activity, which leads to the differential regulation of synaptic transmission by different gliotransmitters.

## Discussion

Present results show that a single astrocyte can release glutamate and ATP/adenosine that potentiate and depress neurotransmitter release, respectively. We show that astrocytes can decode neuronal activity and release glutamate and ATP/adenosine depending on the neuronal activity. That is, relatively short or low frequency stimulation patterns induce glutamate release from astrocytes resulting in a potentiation of the probability of neurotransmitter release; while prolonged or relatively high frequency stimulus also induce ATP/adenosine release from astrocytes, leading to a depression of synaptic transmission (*Figure 7*). These results show that astrocytes decode neuronal activity and tune their response differentially modulating synaptic transmission.

A great amount of data have been reported in the last two decades showing the implication of astrocytes in synaptic modulation, through different mechanisms involving different gliotransmitters (*Navarrete and Araque, 2010*; *Perea et al., 2016*; *Perea and Araque, 2007*; *Min and Nevian, 2012*; *Henneberger et al., 2010*; *Gómez-Gonzalo et al., 2015*; *Navarrete et al., 2012*; *Pascual et al., 2005*; *Serrano et al., 2006*; *Takata et al., 2011*). In the hippocampus, astrocytes are well known to release glutamate, ATP, GABA, and D-serine (*Panatier et al., 2011*; *Perea et al., 2016*; *Henneberger et al., 2010*; *Navarrete and Araque, 2008*; *Pascual et al., 2005*; *Serrano et al., 2006*; *Le Meur et al., 2012*); however, it remained unknown whether a single astrocyte can release all of them or, on the contrary, there are specialized astrocyte subtypes that release each gliotransmitter. Present results show that astrocyte Ca²⁺ elevations induced by endogenous stimuli (GABA released during interneuron activity) lead to a glutamate- and ATP/adenosine-mediated biphasic modulation of neurotransmitter release, an effect that was mimicked by the artificial but direct and selective activation of a single astrocyte with Ca²⁺ uncaging, indicating that both

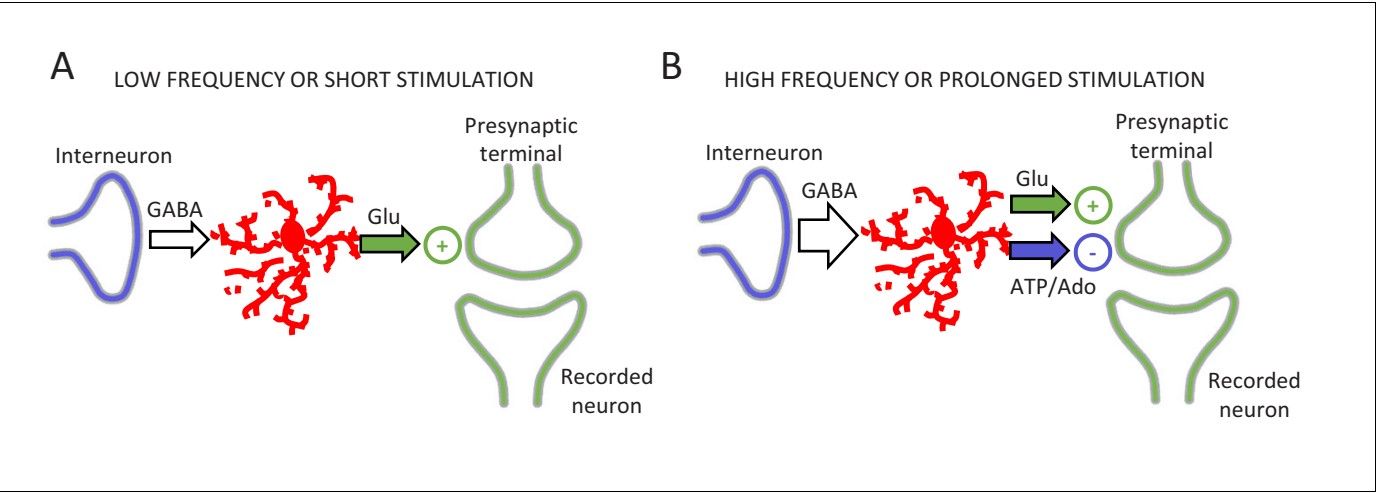

**Figure 7.** Schematic of the signaling pathways involved in gliotransmitter release. (**A**) Low frequency or short interneuron stimulation induces glutamate release from astrocytes leading to a potentiation of neurotransmitter release. (**B**) High frequency or prolonged interneuron stimulation induces glutamate and ATP/adenosine release from astrocytes leading to a synaptic potentiation followed by a synaptic depression.
DOI: https://doi.org/10.7554/eLife.32237.014

gliotransmitters can be released from a single astrocyte. These results suggest that the ability of releasing different gliotransmitters is not diversified in different astrocyte subpopulations, but that single hippocampal astrocytes have the ability to release both gliotransmitters.

Present results were obtained using the minimal stimulation approach, which is considered to allow monitoring the activity of single or very few synapses based on the relatively low amplitude and stability of the synaptic responses (*Navarrete and Araque, 2010*; *Mariotti et al., 2018*; *Chen et al., 2013*; *Isaac et al., 1996*; *Raastad, 1995*). However, the inherent limitation of the approach remained uncertain whether single or few synapses were activated. Nevertheless, the fact that the synaptic potency remained unchanged throughout the experiments indicates that the same synapse or synapses were monitored.

Our results show that the synaptic effects of astrocytic glutamate and ATP/adenosine have different dynamics that lead to a biphasic regulation. Whether this temporal difference is due to different temporal properties of the gliotransmitter release or the upstream or downstream signaling mechanisms requires further investigation. For example, it could be feasible that the cellular machinery responsible for glutamate release is faster than for ATP/adenosine release. Alternatively, if both gliotransmitters are co-released, the buildup of the extracellular concentration of the gliotransmitter, the diffusion properties of the gliotransmitter to reach the appropriate receptors, the location of neuronal receptors in relation to the gliotransmitter source, or the neuronal receptor affinity, are important elements that may account for the different temporal effects. Nevertheless, regardless the actual cellular mechanism, present results indicate that different gliotransmitters can be released from a single astrocyte, but with different temporal effects on synaptic activity and under the control of the neuronal activity that signal to astrocytes.

*Figure 6* shows that astrocytes decode the interneuron activity tuning the release of gliotransmitters. Indeed, different duration or frequency of interneuron activity result in different astrocyte-mediated synaptic modulation. While relatively short stimuli only induced glutamate-mediated potentiation, sustained neuronal activity induced the biphasic synaptic modulation mediated by the initial glutamate-mediated component followed by the ATP/adenosine- phenomenon, suggesting that two temporal thresholds might exist, one for glutamate and another for ATP/adenosine effects. Likewise, while relatively low firing frequencies induces glutamate-mediated potentiation of neurotransmission, relatively high firing frequencies induce glutamate- and ATP/adenosine-mediated biphasic modulatory phenomenon. These results show a relationship between the stimulus frequency and gliotransmission, where glutamate effects occurred at low and high frequency stimuli while ATP/adenosine effects occur only at higher frequencies. Moreover, a slower time course of the biphasic synaptic modulation occurred at higher firing frequencies, indicating that when the neuronal activity

is high the effect of the gliotransmission is longer lasting. Therefore, two different intensity thresholds also exist for glutamate and ATP/adenosine effects. As indicated above, whether these temporal and intensity thresholds are due to different gliotransmitter release mechanisms or to downstream signaling mechanisms requires further studies.

Interestingly, the GABA$_B$R-mediated biphasic synaptic regulatory phenomenon obtained with either interneuron depolarization or with direct baclofen application showed a slower time course than the responses obtained from neuronal ionotropic receptors. This is not surprising considering that this biphasic regulatory phenomenon requires astrocytic GABA$_B$R activation, astrocyte Ca$^{2+}$ signaling in astrocytes and astrocytic glutamate and ATP/adenosine release. In fact, it is known that astrocyte Ca$^{2+}$ signaling occurs in a slow and prolonged manner (*Volterra et al., 2014*; *Araque et al., 2014*; *Rusakov, 2015*) probably due to a different expression of GABA$_B$Rs by distinct astrocyte; to the fact that astrocyte responses require the build-up of extracellular GABA released from synaptic terminals, which may not reach all the monitored astrocytes; or the number of steps involved in activation of G protein-mediated signaling pathways and intracellular Ca$^{2+}$ mobilization.

Present results indicate that synaptic depression is mediated by activation of adenosine A$_1$ receptors. While adenosine can be released through Ca$^{2+}$-independent mechanisms (*Martín et al., 2007*; *Dale et al., 2000*), ATP is released in a Ca$^{2+}$-dependent manner (*Panatier et al., 2011*; *Serrano et al., 2006*). Whether adenosine was directly released by astrocytes or was derived from ATP requires further studies out of the scope of the present work. Nevertheless, the dependence of the synaptic effects on astrocyte Ca$^{2+}$ suggests that the gliotransmitter released is ATP. Indeed, the synaptic depression is absent in IP$_3$R2-null mice, in which G protein mediated Ca$^{2+}$ signal is impaired in astrocytes (*Figure 3J*), and astrocyte Ca$^{2+}$ uncaging is sufficient to induce the synaptic depression (*Figure 5*).

Besides glutamate and ATP/adenosine, other gliotransmitters, such as D-serine, have been shown to be released by hippocampal astrocytes (*Henneberger et al., 2010*; *Shigetomi et al., 2013*). While our study focused on glutamate and ATP/adenosine signaling, further studies are needed to determine whether D-serine can also be released by the same astrocytes, which seems likely because present results indicate that single astrocytes are competent to release different gliotransmitters.

In summary, present results show that single astrocytes release different gliotransmitters depending on neuronal activity, by decoding neuronal activity and adjusting their response to distinct physiological stimuli to differentially regulate synaptic transmission. Therefore, astrocytes decipher and integrate synaptic information to differentially regulate synaptic activity through the release of different gliotransmitters depending on the stimulating neuronal activity.

## Materials and methods

### Ethics statement

All of the procedures for handling and sacrificing animals were approved by the University of Minnesota Institutional Animal Care and Use Committee (IACUC) in compliance with the National Institutes of Health guidelines for the care and use of laboratory animals (#1701A34507).

### Mouse lines

Mice were housed under 12/12 hr light/dark cycle and up to five animals per cage. Hippocampal slices were obtained from male and female C57BL/6J (12–21 days old), IP$_3$R2-null (12–21 days old) (*Li et al., 2005*; *Zimmer et al., 1999*), Pvalb-Cre/Ai9-rcl-tdTomato (PV$^+$, 12–21 days old; JAX #008069/#007909), GFAP-CB1-null (12–20 week old) (*Han et al., 2012*; *Martin-Fernandez et al., 2017*) and GLAST-GABA$_B$-null (12–20 week old) (*Perea et al., 2016*). IP$_3$R2-null mice were generously donated by Dr. J. Chen. To generate astroglial Cnr1$^{-/-}$ (GFAP-CB1-null) mice carrying the ''floxed'' Cnr1 gene (Cn1f/f) were crossed with Gfap-CreERT2 mice, using a three step backcrossing procedure we obtained Cnr1f/f;Gfap-CreERT2 (GFAP-CB1-null mice). Astroglial GABA$_B$-null was generated by crossbreeding Gabbr1fl/fl (MGI:3512742) carrying the 'floxed' Gabbr1 gene (*Haller et al., 2004*) with Glast-CreERT2 knock-in mice (MGI:3830051) (*Mori et al., 2006*). R26-lsl-GCaMP3 mice (JAX #014538) (*Paukert et al., 2014*) were crossbred to astroglial Gabbr1$^{-/-}$ mice to express the genetically encoded Ca$^{2+}$ indicator GCaMP3 specifically in astrocytes. As CreERT2 protein is inactive in the absence of tamoxifen treatment, deletion of the Cnr1 or Gabb1 genes was

obtained in adult mice (8 weeks) by eight daily injections of tamoxifen (1 mg, i.p.), dissolved in 90% sunflower oil, 10% ethanol to a final concentration of 10 mg/ml. The animals were used at least 4 weeks after tamoxifen treatment. *Cnr1f/f* and *Gabbr1fl/fl* mice injected with tamoxifen were used as WT controls for GFAP-CB1-null and GABA$_B$-null mice, respectively.

## Hippocampal slice preparation

Animals were anaesthetized and decapitated. The brain was rapidly removed and placed in ice-cold artificial cerebrospinal fluid (ACSF). Slices 350 µm thick were made and incubated (>30 min) at room temperature in ACSF containing (in mM): NaCl 124, KCl 5, NaH$_2$PO$_4$ 1.25, MgSO$_4$ 2, NaHCO$_3$ 26, CaCl$_2$ 2, and glucose 10, and was gassed with 95% O$_2$/5% CO$_2$ (pH = 7.3–7.4). Slices were transferred to an immersion recording chamber, superfused at 2 ml/min with gassed ACSF and visualized under an Olympus BX50WI microscope (Olympus Optical, Japan).

## Electrophysiology

Electrophysiological recordings from CA1 pyramidal neurons and *Stratum radiatum* (*S. R.*) interneurons were made in whole-cell configuration of the patch-clamp technique. Patch electrodes had resistances of 3–10 MΩ when filled with the internal solution containing (in mM): KMeSO$_4$ 135, KCl 10, HEPES 10, NaCl 5, ATP-Mg$^{+2}$ 2.5, and GTP-Na$^+$ 0.3 (pH = 7.3). In some experiments astrocytes were patched with 4–9 MΩ electrodes filled with an intracellular solution containing (in mM): KMeSO$_4$ 100, KCl 50, HEPES-K 10, and ATP-Na$^{+2}$ 4 (pH = 7.3). Recordings were obtained with PC-ONE amplifiers (Dagan Instruments, MN, US). Membrane potential was held at −70 mV for neurons and −80 mV for astrocytes. Series and input resistances were monitored throughout the experiment using a −5 mV pulse. Cells were discarded when series and input resistances changed >20%. Signals were fed to a Pentium-based PC through a DigiData 1440A interface board. Signals were filtered at 1 KHz and acquired at 10 KHz sampling rate. The pCLAMP 10.4 (Axon instruments) software was used for stimulus generation, data display, acquisition and storage. Inhibitory postsynaptic potentials (IPSCs) were isolated in the presence of CNQX (20 µM) and AP5 (50 µM) to block AMPA and NMDA receptors, respectively.

## Synaptic stimulation

Theta capillaries filled with ACSF were used for bipolar stimulation and placed in the *S. R.* to stimulate Schaffer collaterals (SC). Paired pulses (2 ms duration with 50 ms interval) were continuously delivered at 0.33 Hz using a stimulator S-910 through an isolation unit. Putative excitatory post-synaptic currents (EPSCs) were recorded from pyramidal neurons. Stimulus intensity (0.1–10 mA) was adjusted to meet the conditions that putatively stimulate single or very few presynaptic fibers (cf. [*Navarrete and Araque, 2010*; *Perea and Araque, 2007*; *Isaac et al., 1996*; *Raastad, 1995*; *Allen and Stevens, 1994*]). Synaptic parameters analyzed were: probability of release (ratio between the number of successes versus total number of stimuli); synaptic efficacy (mean peak amplitude of all responses including successes and failures) and synaptic potency (mean peak amplitude of the successes, excluding failures) (*Navarrete and Araque, 2010*; *Perea and Araque, 2007*; *Gómez-Gonzalo et al., 2015*; *Navarrete et al., 2012*). A response was considered a failure if the amplitude of the current was <3 times the standard deviation of the baseline current (0.7–2.3 pA) and was verified by visual inspection.

The high frequency stimulation (HFS) paradigm consisted in a Tetanic stimulation (3 trains at 100 Hz for 1 s delivered every 30 s) delivered on independent SC pathways. The independence of the SC pathways was confirmed by stimulating both pathways with time intervals of 50 ms and observing no cross-facilitation of the EPSCs.

## Neuronal stimulation and data analysis

During the neuronal depolarization (ND) protocol one pyramidal neuron was depolarized to 0 mV for 5 s to stimulate endocannabinoid release (*Navarrete and Araque, 2010*; *Navarrete and Araque, 2008*; *Chevaleyre and Castillo, 2004*; *Kreitzer and Regehr, 2001*; *Wilson and Nicoll, 2001*; *Ohno-Shosaku et al., 2002*). During the interneuron depolarization train paradigm, the interneuron was recorded in current-clamp mode and depolarized so it fired action potentials at 1, 10, 20, 30 or 50 Hz for 90 s. Interneurons were also depolarized at 20 Hz for 100 ms, 1, 30, and 60 s. To determine

the interneuron firing frequency, interneurons were depolarized for 13 ms and repolarized so they fired a single action potential in every depolarization (*Figure 3A*). During the interneuron depolarizing train, synaptic activity was continuously monitored at 0.33 Hz in the pyramidal neuron.

To illustrate the time course of the effects of the stimuli, synaptic parameters were grouped in 30 or 60 s bins. To determine synaptic changes, mean EPSCs (n = 60 EPSCs) recorded 3 min before the stimulus (basal) were compared with mean EPSCs (n = 20 EPSCs) after the stimulus (for potentiation) or after the initial transient synaptic potentiation subsided (for depression). The synaptic potentiation or depression was determined to occur in individual synapses when the probability of release determined after the stimuli changed more than two standard deviations from the baseline. *Figure 2F* and *Figure 4—figure supplement 1A* shows the percentage of synapses undergoing potentiation, depression, both or no change. Synapses that did not show synaptic regulation in control conditions were not further considered. This selection criteria was necessary because pulling together results from regulated or unregulated synapses would result in the concealment of the effects (*Lines et al., 2017*). Considering that the regulation of single or few synapses was analyzed, it is not unexpected that in some experiments the stimulus was inefficient to regulate some synapses, perhaps simply due to the absence of functional connectivity between the stimulated astrocyte and the recorded synapse. Hence, for pharmacology experiments and experiments using transgenic mice, a relatively large sample size is used, thus the absence of the phenomena can be assumed by never being observed in the recorded population.

## Ca²⁺ imaging

$Ca^{2+}$ levels in astrocytes located in the *S. R.* of the CA1 region of the hippocampus were monitored by fluorescence microscopy using the $Ca^{2+}$ indicator fluo-4-AM. Astrocytes where loaded with fluo-4-AM by incubating the slices with fluo-4-AM (2 μM and 0.01% of pluronic) for 45–60 min at room temperature (*Parri et al., 2001*; *Perea and Araque, 2005*; *Kang et al., 1998*; *Araque et al., 2002*; *Nett et al., 2002*). Astrocytes were imaged using a CCD camera (Hammamatsu, Japan). Cells were illuminated during 100 ms with a LED at 490 nm (Prior Scientifics, MA, US), and images were acquired every 2 s. The LED and the CCD camera were controlled and synchronized by the Meta-Morph software (Molecular devices). ImageJ software (NIH) was used for quantitative epifluorescence measurements. $Ca^{2+}$ variations recorded at the soma of the cells were estimated as changes of the fluorescence signal over baseline ($\Delta F/F_0$), and cells were considered to show a $Ca^{2+}$ event when $\Delta F/F_0$ increased two times the standard deviation of the baseline.

The astrocyte $Ca^{2+}$ signal was quantified from the astrocyte $Ca^{2+}$ event probability, i. e., the probability of occurrence of a $Ca^{2+}$ elevation, which was calculated from the number of $Ca^{2+}$ elevations. The time of occurrence of a $Ca^{2+}$ event was determined at the onset of the $Ca^{2+}$ event. Astrocyte $Ca^{2+}$ events obtained from 5 to 25 astrocytes in the field of view during the recording period were grouped in 5 s bins. For each astrocyte analyzed, values of 0 and 1 were assigned for bins showing either no response or a $Ca^{2+}$ event, respectively. To test the effects on $Ca^{2+}$ event probability under different conditions, the basal (15 s before the stimulus) and maximum $Ca^{2+}$ event probability (15 s after stimulus) were averaged and compared.

## Ca²⁺uncaging by UV-flash photolysis

Single astrocytes were recorded with patch pipettes filled with the internal solution containing 5 mM NP-EGTA, 160 μM fluo-4-AM to -monitor $Ca^{2+}$ levels and GDPßS (10 mM). $Ca^{2+}$ uncaging was achieved by illuminating the slice with UV light during 90 s using a 430–480 nm LED (Prior Scientifics, MA, US).

## Biocytin labeling

Single astrocytes were recorded with patch pipettes and filled with internal solution containing 0.5% biocytin. Slices were fixed in 4% PFA in 0.1 PBS (pH 7.4) at 4°C. Slices were washed three times in 1xPBS (10 min each). To visualize biocytin slices were incubated with Alexa488-Streptavidin (RRID: AB_2315383; 1:500) for 48 hr at 4°C. Slices were finally washed for 3 times with 1xPBS (10 min each) and mounted with Vectashield mounting media (Vector laboratories).

## Statistics

Data are expressed as mean $\pm$ standard error of the mean (SEM). Results were compared using a two-tailed, paired Student's $t$-test ($\alpha = 0.05$). Statistical differences were established with $p < 0.05$ (*), $p < 0.01$ (**) and $p < 0.001$ (***). Blind experiments were not performed in the study but the same criteria were applied to all experimental groups. Randomization was not employed. Sample size was based on values previously found sufficient to detect significant changes in hippocampal synaptic parameters in past studies performed in the lab.

## Drugs and chemicals

(S)-(+)-$\alpha$-Amino-4-carboxy-2-methylbenzeneacetic acid (LY367385), ($2S$)−3-[($1S$)−1-(3,4-Dichlorophenyl)ethyl]amino-2-hydroxypropyl](phenylmethyl)phosphonic acid hydrochloride (CGP 55845) were purchased from Tocris Cookson (Bristol, UK); Fluo-4-AM, Fluo-4 cell impermeant and o-nitrophenyl EGTA, tetrapotassium salt (NP-EGTA) from Molecular Probes (Eugene, OR); and Picrotoxin from Indofine Chemical Company (Hillsborough, NJ). All other drugs were purchased from Sigma.

## Acknowledgements

We thank R Quintana and S Jamison for technical help; Dr. J Chen (UCSD, USA) for providing IP$_3$R2-null mice; Dr. G Marsicano for providing the GFAP-CB1-null; Dr. F Kirchhoff, Dr. D E Bergles and Dr. A Agarwal for providing the astroglial GABA$_B$-null and GCaMP3 mice; and Dr. R Gomez, Dr. A Diez, M Martin-Fernandez, C Durkee, J Lines, M Corkrum and A Ferro for helpful suggestions. This work was supported by NIH-NINDS (R01NS097312-01) and Human Frontier Science Program (Research Grant RGP0036/2014) to AA. The authors declare no financial competing interests.

## Additional information

### Funding

| Funder | Grant reference number | Author |
| --- | --- | --- |
| Human Frontier Science Program | RGP0036/2014 | Alfonso Araque |
| National Institute of Neurological Disorders and Stroke | R01NS097312-01 | Alfonso Araque |

The funders had no role in study design, data collection and interpretation, or the decision to submit the work for publication.

### Author contributions

Ana Covelo, Designed the study, Performed experiments, Analyzed data, Wrote the paper, Conception and design, Acquisition of data, Analysis and interpretation of data, Drafting or revising the article; Alfonso Araque, Designed the study, Wrote the paper, Conception and design, Analysis and interpretation of data, Drafting or revising the article

### Author ORCIDs

Ana Covelo https://orcid.org/0000-0002-9201-4703
Alfonso Araque https://orcid.org/0000-0003-3840-1144

### Ethics

Animal experimentation: All of the procedures for handling and sacrificing animals were approved by the University of Minnesota Institutional Animal Care and Use Committee (IACUC) in compliance with the National Institutes of Health guidelines for the care and use of laboratory animals (#1701A34507).

### Decision letter and Author response

Decision letter https://doi.org/10.7554/eLife.32237.017

Author response https://doi.org/10.7554/eLife.32237.018

## Additional files

### Supplementary files
• Transparent reporting form
DOI: https://doi.org/10.7554/eLife.32237.015

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
