## [Decision Letter]

[Editors’ note: a previous version of this study was rejected after peer review, but the authors submitted for reconsideration. The first decision letter after peer review is shown below.]

Thank you for submitting your work entitled "Neuronal activity determines distinct gliotransmitter release from a single astrocyte" for consideration by *eLife*. Your article has been evaluated by a Senior Editor and three reviewers, one of whom is a member of our Board of Reviewing Editors. The reviewers have opted to remain anonymous. Our decision has been reached after consultation between the reviewers. Based on these discussions and the individual reviews below, we regret to inform you that your work will not be considered further for publication in *eLife*.

Summary:

The reviewers agreed that determining the mechanisms that enable astrocytes to influence synaptic transmission and plasticity is critically important and that the strategies you employed to explore the modulatory output of single astrocytes were elegant. The idea that the release of ATP and glutamate are induced by different stimuli was also seen as an important discovery. However, there was consensus that the complexity of the system creates challenges for studies based on pharmacological manipulations and the reviewers thought that targeted gene deletion from astrocytes is necessary to provide a definitive test of the model. In addition, there were concerns raised about the manner of data analysis, the reliance on failures, rather than paired pre- and post-synaptic recording, and the use of non-physiological stimuli in some experiments. Together, the reviewers thought that these issues reduced the overall impact of the study.

Reviewer #1:

This report by Covelo and Araque describes a technically challenging set of experiments in hippocampal slices to explore the involvement of astrocyte calcium signaling in the regulation of excitatory Schaffer collateral commissural synapses in the CA1 region. Gliotransmitter release and the role of astrocytes in synaptic modulation continue to be intensely debated within the field, with substantial evidence accumulated both for and against this hypothesis. Developing a minimal system in which key players can be adequately controlled and manipulated would help advance this field. The studies in this report rely on "minimal stimulation" to isolate a small number of synapses, providing a more rigorous evaluation of this phenomenon than can be achieved with large scale extracellular stimulation. Using paired recordings, calcium imaging and calcium uncaging in astrocytes, and manipulations of PV interneurons, the authors derive a complex model to describe how individual astrocytes can release either glutamate or ATP, depending on the duration and amplitude of the stimulus, in response to GABA release from interneurons. Of note, the study would benefit from a summary diagram outlining the major pathways involved; at times, I found it difficult to follow the manipulations and the assumptions that were made. As with any complex system, there are many possible pathways that could be responsible for the observed effects, particularly because the time course of the events is slow – the stimulus duration, calcium rises in astrocytes and the potentiation – and the receptors targeted by the pharmacological manipulations are expressed by multiple cell types. Without greater control of receptors on specific cells other interpretations are possible. Accordingly, there are several concerns that should be addressed to ensure that the authors' interpretation of these elegant manipulations is correct.

1) The studies build upon the author's prior work indicating that pyramidal neuron depolarization elicits cannabinoid release that induces synaptic potentiation. It was unclear to me from the experiments performed whether astrocytes were the site of action or interneurons. Clearly the latter have been shown to be a target in this region of the hippocampus, as this is where DSI was first demonstrated. Some evidence of cannabinoid receptor involvement in the phenomena described here would be helpful to establish the initial conditions. To counter possible complications from neuronal CB1 receptors, the authors should consider analyzing astrocyte-specific CB1 receptor knockout mice. If the central hypothesis is correct, this should abolish both potentiation and depression of SC EPSCs.

2) The studies rely on minimal stimulation to isolate one or a few synapses. This approach has been used previously in this region. Although not as powerful as paired recordings, the hippocampal system is not amenable to this approach, as it is difficult to find pairs of synaptically coupled CA3-CA1 neurons in slices. Accordingly, there are many caveats with minimal stimulation, as it is certain that many axons are stimulated (although one or a few may be connected to the cell being recorded from). In order to perform failure analysis and determine release probability (used in much of the quantification throughout the study), it is necessary to be able to resolve successes from failures. In this region, it is very challenging to do so, because EPSC amplitudes can be very small (plots of mEPSC amplitudes do not show a clear separation from the noise and events), creating challenges for this type of analysis. In Figure 1, the large separation between successes and failures is concerning, as this degree of separation is not typical for these synapses. The sharp boundary of the smallest events suggests that anything below 5pA was cut off, creating an artifactual separation that enhances the ability to distinguish successes and failures. The issue should be addressed.

3) I am confused about the manner of data representation in several of the experiments. The authors report that 31% of "single synapses" showed modulation. Do the p values represent all experiments or only those that showed modulation? This data selection should be avoided unless there is a clear mechanistic argument to support this heterogeneity. There are similar issues for the calcium signals from astrocytes. Here it is important to define what is meant by the metric "calcium event probability". What accounts for the variability in responsiveness (certain cells respond) and variability in calcium increases? If baclofen were applied acutely, does it elicit similar responses (calcium increases in astrocytes and biphasic changes in Pr)? What accounts for the delay in response of astrocytes – that is, if this is a simple mobilization of IP3 following GABA_B_ receptor activation, why is the onset so variable?

4) A key experiment suggesting the involvement of astrocytes involves analysis of synaptic potentiation and depression in IP_3_R2 ko mice. The authors should perform a control experiment to show that calcium uncaging in astrocytes in IP_3_R2 ko mice leads to the same biphasic change in Pr.

5) Because GABA_B_ receptors are expressed by multiple cell types, the authors should consider performing these experiments in astrocyte specific conditional knockout mice. Such experiments would lend stronger support for the proposed model of neuromodulation.

*Reviewer #2:*

Studies on the ability of astrocytes to regulate ongoing synaptic transmission have focused on release of an individual gliotransmitter from astrocytes and one type of regulation such as potentiation or depression of presynaptic release probability. This paper addresses whether one astrocyte can induce both potentiation and inhibition of presynaptic release at the same synapse, via the release of different transmitters. The study shows that the effect of astrocytes on synapses is time and neuronal activity-dependent. Astrocytes first potentiate presynaptic release via glutamate acting on mGluRs. After a delay presynaptic release is inhibited via adenosine acting on A1 receptors. The duration and intensity of neuronal activation regulates the astrocyte/synapse response, with low intensity activation only causing synaptic potentiation, and high intensity stimulation causing both.

Showing that one astrocyte can have both effects on the same synapse is interesting, but it is hard to interpret the physiological meaning of the results the way they are currently presented. The incorporation of a model figure depicting the proposed circuit connections between the principal neurons, interneurons and astrocytes would help in interpretation, as would addressing the following questions about whether the neuronal stimulation parameters used correlate with endogenous firing properties of the neurons. These points do not need new experiments to address them.

In Figure 3 activation of the interneuron causes potentiation of an excitatory synapse (via the astrocyte). Is this known/expected to happen?

Are the stimuli used for interneuron activation within the range of what is normally recorded in the hippocampus (20Hz for 90sec)?

Is there evidence that astrocytes undergo prolonged somatic calcium rises (seconds)? The effects on synapses on the whole seem to be on a slow timescale (minutes).

Does the size of the astrocyte calcium response correlate with the outcome at the synapse, i.e. just potentiation, or potentiation followed by depression? In Figure 3 the astrocyte calcium increases at the start of interneuron stimulation but has decayed back to baseline after 25sec even though the neuronal stimulus lasts for 90sec – is this a saturating effect?

*Reviewer #3:*

The author (and others in the astrocyte field) has developed a consistent story over the past years describing a link between astrocyte calcium signals and alteration of neuronal synaptic responses. Based on pharmacological blockers the scenario is that CB1 receptor stimulation of astrocytes triggers calcium signals that in turn trigger glutamate release from astrocyte that enhances synaptic transmission. This excitatory response is transient and is mediated by mGluR1 receptors. The second pathway is proposed to be due to GABA_B_-mediated calcium transients in astrocytes (activated by GABA release from interneurons) causes ATP release from astrocytes and subsequent depression of neuronal activity due to adenosine formation from degradation of ATP. In the context of these previous studies that were reported in publications from this lab and others, the authors have designed experiments to determine whether one astrocyte can have similar actions of excitation and inhibition. They provide evidence supporting their conclusion that calcium signals in one identified astrocyte can lead to both types of synaptic modifications. However the authors have no data that ATP is released from astrocytes under these circumstances. Their data show that there is an adenosine-mediated depression of synaptic responses and the role for ATP release is inferred from other studies.

The authors describe the results using several parameters such as potency, probability of release and synaptic efficacy. These are defined as "probability of release (ratio between the number of successes versus total number of stimuli); synaptic efficacy (mean peak amplitude of all responses) and synaptic potency (mean peak amplitude of the successes)". From the data presented in Figure 1 it appears that the impact on synaptic release is to change the failure rate for evoked epscs using this experimental paradigm. The authors describe the potency presumably as the amplitude of the evoked epscs excluding the failures. This part of the data presentation is confusing and should be clarified in the figure legend as well as the brief description in the Materials and methods. For example in Figure 2 the actual evoked responses are shown in the top traces and the average responses are shown in the lower traces. These averages must also include the failures as well as the evoked synaptic currents. For example the response in ND appear to have several responses with smaller amplitude and the responses during the HFS appear to be identical to the basal responses. This could be important because of the implication of the mechanism. The authors conclude that the release of transmitter at synapses is inhibited. However the impact may be the inhibition of the propagation of action potentials or failure of spikes to invade the presynaptic terminal instead of depression of synaptic release mechanisms.

"These results indicate that interneuron-evoked stimulation of astrocytes leads to the potentiation and depression of synaptic transmission." This conclusion depends on the assumption that IP_3_R2 receptors are only in astrocytes and this assumption should be clearly stated. It is certainly true that IP_3_R2 are important for the calcium signals linked to Gq coupled GPCRs. The references cited here all deal with the loss of calcium signaling in astrocytes in the IP_3_R2 knockout mouse. However it is possible that there are IP_3_R2 receptors in some neurons or specific neuronal compartments. The conclusion that this is solely due to astrocytes would require an astrocyte selective knockout of IP_3_R2 and I am not aware of this evidence (although I could be wrong). The authors should also be aware of several studies showing that calcium signaling stills occurs in fine processes of astrocytes in the IP_3_R2^-/-^ transgenic mouse.

The authors have provided intriguing evidence that the pattern of action potential firing in interneurons will lead to the synaptic enhancement or synaptic depression. This is a remarkable finding that the authors have convincingly shown using a very impressive strategy involving paired recordings from identified cells in brain slices. The initial comparison in Figure 2 for neuronal depolarization (CB1 mediated enhancement) versus high frequency stimulation (GABAG mediated) seems puzzling. The authors could describe more clearly the rationale for these initial experiments then the progression to the interneuron specific responses. In particular why was no initial enhancement of synaptic responses observed in Figure 2, compared to the enhancement in Figure 3?

[Editors’ note: what now follows is the decision letter after the authors submitted for further consideration.]

Thank you for resubmitting your work entitled "Neuronal activity determines distinct gliotransmitter release from a single astrocyte" for further consideration at *eLife*. Your revised article has been favorably evaluated by Gary Westbrook (Senior Editor) and three reviewers, one of whom is a member of the Board of Reviewing Editors. The manuscript has been improved but there are some remaining issues that need to be addressed before we can make a decision on acceptance, as outlined below:

*Reviewer #1:*

The authors have conducted many additional experiments to address the concerns raised and have clarified some issues that were confusing. As a result, the manuscript is much improved. However, I have several additional comments.

1) The bar graphs analyzing changes in release probability, potency, etc. show effects that were present during the initial time after the manipulation. Please indicate the frequency of test stimulation after the inducing stimulus (I might have missed it, but wasn't able to find this in the Materials and methods). Given that repetitive stimulation is required to assess these parameters, it is important to indicate how many responses were used to calculate these averages.

2) The inclusion of astrocyte-specific deletion experiments for CB1Rs and GABABRs is a terrific addition to the study. However, the authors need to provide evidence that this genetic manipulation was effective by directly assessing the effects of a CB1 agonist and GABA/baclofen on Ca changes in astrocytes, similar to experiments shown in Figure 3—figure supplement 1. It should be noted that both manipulations (using GFAP-CreER) may lead to deletion of CB1 receptors from radial glia and thus eventually dentate granule neurons, with unknown consequences for the tri-synaptic hippocampal circuit.

3) I am not convinced by either the data or the argumentation that the responses recorded represent exclusively single synapses. Although the author correctly indicates that this approach has been used previously, there remains considerable uncertainty that is inherent to the approach. Description of responses as single synapses should formally be reserved for paired recordings in which the number of functional connections can be directly established (typically through post-hoc immunolabeling). For example, the representative example data shown in Figure 1 highlights the difficulty in defining what is a single synapse response, as the response amplitude clearly varies continuously with no clear separation between what are deemed successes and failures (by coloration). The statements surrounding single synapses should be softened to allow for this uncertainty.

4) I remain concerned about the use of data selection in the manuscript. Data should not be excluded simply because the cells did not respond as expected. Exclusion criteria should be clearly defined, for example, if the authors were to show that astrocytes varied in their expression of receptors, by challenging cells at the end of the experiment with a selective agonist. Regarding the source of variability, please indicate what proportion of astrocytes exhibited Ca increases in response to baclofen application.

*Reviewer #2:*

The authors have addressed the questions I raised in the first round of review in a satisfactory manner. The other reviewers raised more substantive concerns. In particular, the point reviewer 1 raised about the ability of minimal stimulation in the CA3-CA1 pathway to be monitoring the same synapse every time seems particularly important for the interpretation of the data – that one synapse is both potentiated and depressed by different gliotransmitters from the same astrocyte. Additionally, the new experiments using tamoxifen-inducible astrocyte-specific KO lines of different receptors don't appear to have any validation in this paper (there are references provided but one is from a different lab, and the other uses recombination in younger animals), so this is something that needs adding, or a more detailed explanation of validation of the models should be provided in the Materials and methods.

*Reviewer #3:*

I am impressed by the additional data that the authors have provided to deal with questions from me and the other reviewers. The additional clarification of several issues such as the description of the measured synaptic parameters and schematic summaries have greatly helped. The inclusion of data from using astroglial CB1R^-/-^ mice, astroglial GABA_B_R^-/-^ mice, IP_3_R2^-/-^ mice are very powerful and convincing additions to their body of evidence pointing to the complexity of astrocyte modulation of synaptic activity. In addition the new experiment to block G protein signalling in astrocytes (by injecting GDPbetaS into astrocytes) to block astrocyte calcium elevations and biphasic synaptic regulation was a clever addition. The inclusion of new genetic knock-down experiments and selective depression of astrocyte signaling provides convincing data and answers the issues raised in the first reviews.

---

## [Author Response]

[Editors’ note: the author responses to the first round of peer review follow.]

Reviewer #1:This report by Covelo and Araque describes a technically challenging set of experiments in hippocampal slices to explore the involvement of astrocyte calcium signaling in the regulation of excitatory Schaffer collateral commissural synapses in the CA1 region. Gliotransmitter release and the role of astrocytes in synaptic modulation continue to be intensely debated within the field, with substantial evidence accumulated both for and against this hypothesis. Developing a minimal system in which key players can be adequately controlled and manipulated would help advance this field. The studies in this report rely on "minimal stimulation" to isolate a small number of synapses, providing a more rigorous evaluation of this phenomenon than can be achieved with large scale extracellular stimulation. Using paired recordings, calcium imaging and calcium uncaging in astrocytes, and manipulations of PV interneurons, the authors derive a complex model to describe how individual astrocytes can release either glutamate or ATP, depending on the duration and amplitude of the stimulus, in response to GABA release from interneurons. Of note, the study would benefit from a summary diagram outlining the major pathways involved; at times, I found it difficult to follow the manipulations and the assumptions that were made. As with any complex system, there are many possible pathways that could be responsible for the observed effects, particularly because the time course of the events is slow – the stimulus duration, calcium rises in astrocytes and the potentiation – and the receptors targeted by the pharmacological manipulations are expressed by multiple cell types. Without greater control of receptors on specific cells other interpretations are possible. Accordingly, there are several concerns that should be addressed to ensure that the authors' interpretation of these elegant manipulations is correct.

We thank the reviewer for the positive comments and suggestions that helped to strengthen the paper. As suggested, summary diagrams outlining the major pathways involved have been included in new Figure 2 and new Figure 7.

1) The studies build upon the author's prior work indicating that pyramidal neuron depolarization elicits cannabinoid release that induces synaptic potentiation. It was unclear to me from the experiments performed whether astrocytes were the site of action or interneurons. Clearly the latter have been shown to be a target in this region of the hippocampus, as this is where DSI was first demonstrated. Some evidence of cannabinoid receptor involvement in the phenomena described here would be helpful to establish the initial conditions. To counter possible complications from neuronal CB1 receptors, the authors should consider analyzing astrocyte-specific CB1 receptor knockout mice. If the central hypothesis is correct, this should abolish both potentiation and depression of SC EPSCs.

We thank the reviewer for the insightful comment. To further clarify the cellular target of endocannabinoids (eCBs), we have performed new experiments using astroglial CB1R^-/-^ mice that specifically lack CB1Rs in astrocytes. In these mice, eCBs released by neuronal depolarization (ND) failed to induce synaptic changes (new Figure 2—figure supplement 1), confirming the hypothesis that eCB-mediated synaptic potentiation requires the activation of CB1Rs in astrocytes (see scheme in new Figure 2). Moreover, in those mice, high-frequency stimulation (HFS) induced a synaptic depression similar to wild type mice (new Figure 2—figure supplement 1), confirming that CB1Rs are not involved in GABA-mediated heterosynaptic depression (see scheme in new Figure 2). We now report these findings: “ND-evoked synaptic potentiation was prevented in the presence of the mGluR_1_ antagonist LY367385 (100 µM; n=4 neurons; p=0.928) (Figure 2) and in the astroglial CB1^-/-^ mice (n=10; p=0.8; Figure 2—figure supplement 1), while the HFS-mediated heterosynaptic depression was blocked by the A_1_ receptor (A_1_R) antagonist CPT (2 µM; n=6 neurons; p=0.362) (Figure 2) but still present in the astroglial CB1^-/-^ mice (n=10; p=0.8; Figure 2—figure supplement 1). These results indicate that a single synapse can be modulated by both glutamate and ATP/adenosine released by astrocytes in response to ND or HFS, respectively.”

Moreover, we have performed additional experiments to test that astrocytes are the site of action of interneurons, using astroglial GABA_B_R^-/-^ mice that lack GABA_B_Rs specifically in astrocytes. In these mice, interneuron stimulation failed to induce changes in both in astrocyte Ca^2+^ levels and synaptic transmission, supporting the original hypothesis that GABA released by interneurons activates astrocytic GABA_B_Rs leading to a biphasic regulation of synaptic transmission. These new results have been included in Figure 3 and in the text: “In addition, experiments performed in a transgenic mouse that lacks GABA_B_ receptors specifically in astrocytes (GABAB^-/-^) (Perea et al., 2016) also failed to induce Ca^2+^ elevations in astrocytes (n=53 astrocytes in n=10 slices; p=0.23; Figure 3) and the biphasic synaptic regulation (n=9 neurons; Pr potentiation p=0.75; Pr depression p=0.66; Figure 3). Taken together, these data indicate that GABAB-mediated astrocyte Ca^2+^ signaling is necessary to induce the interneuron-evoked potentiation and depression of synaptic transmission.”

2) The studies rely on minimal stimulation to isolate one or a few synapses. This approach has been used previously in this region. Although not as powerful as paired recordings, the hippocampal system is not amenable to this approach, as it is difficult to find pairs of synaptically coupled CA3-CA1 neurons in slices. Accordingly, there are many caveats with minimal stimulation, as it is certain that many axons are stimulated (although one or a few may be connected to the cell being recorded from). In order to perform failure analysis and determine release probability (used in much of the quantification throughout the study), it is necessary to be able to resolve successes from failures. In this region, it is very challenging to do so, because EPSC amplitudes can be very small (plots of mEPSC amplitudes do not show a clear separation from the noise and events), creating challenges for this type of analysis. In Figure 1, the large separation between successes and failures is concerning, as this degree of separation is not typical for these synapses. The sharp boundary of the smallest events suggests that anything below 5pA was cut off, creating an artifactual separation that enhances the ability to distinguish successes and failures. The issue should be addressed.

We agree with the reviewer that paired recordings is not experimentally feasible. We also agree that minimal stimulation is demanding. Nevertheless, this technique has been successfully used to study single-synapse activity by many laboratories (e.g., Isaac et al., 1996; Rastaad, 1995; Allen and Stevens, 1994), including ours (Perea and Araque 2007; Navarrete and Araque, 2010; Gomez-Gonzalo et al., Cer cortex 2015). For clarity, in the original manuscript, failures were assigned a zero value. To address the reviewer’s concern, we have now included the actual values of the failures as well as the histogram of failures and successes in Figure 1, showing that, in the vast majority of cases, failures and successes can be confidentially discerned (cf. Navarrete and Araque, 2010). Moreover, the analysis performed has been also clarified in the Materials and methods section: “a response was considered a failure if the amplitude of the current was < 3 times the standard deviation of the baseline current (0.7–2.3 pA) and was verified by visual inspection.”

3) I am confused about the manner of data representation in several of the experiments. The authors report that 31% of "single synapses" showed modulation. Do the p values represent all experiments or only those that showed modulation? This data selection should be avoided unless there is a clear mechanistic argument to support this heterogeneity. There are similar issues for the calcium signals from astrocytes. Here it is important to define what is meant by the metric "calcium event probability". What accounts for the variability in responsiveness (certain cells respond) and variability in calcium increases? If baclofen were applied acutely, does it elicit similar responses (calcium increases in astrocytes and biphasic changes in Pr)? What accounts for the delay in response of astrocytes – that is, if this is a simple mobilization of IP3 following GABA_B_ receptor activation, why is the onset so variable?

We apologize for the confusion that was due to our lack of clarity in the previous version. The presence of synaptic changes was assessed for each individual synapse comparing values before and after stimulation. Because not all the single synapses recorded are expected to undergo synaptic regulation by the stimulated astrocyte (cf. Pere and Araque, 2007; Navarrete and Araque, 2010), we believe that data selection is necessary; otherwise, the phenomenon would be concealed when pooling together regulated and unregulated synapses. Therefore, the P value correspond to synapses that showed regulation. We have clarified the issue in the Materials and methods section: “the synaptic potentiation or depression was determined to occur in individual synapses when the probability of release determined after the stimuli changed more than 2 standard deviations from the baseline.”

We also apologize for not being more descriptive regarding the calcium event probability, because we thought it was sufficient to refer to the definition provided in our previous publications (e.g., Navarrete and Araque, 2010; Gomez-Gonzalo et al., Cer cortex 2015). We have now provided a more detailed description in the Materials and method section: “The astrocyte Ca^2+^ signal was quantified from the astrocyte Ca^2+^ event probability, i. e., the probability of occurrence of a Ca^2+^ elevation, which was calculated from the number of Ca^2+^ elevations. […] To test the effects on Ca^2+^ event probability under different conditions, the basal (15 s before the stimulus) and maximum Ca^2+^ event probability (15 s after stimulus) were averaged and compared.”

The source of the variability of the astrocyte responsiveness is a very interesting question. To clarify this issue, we have performed the experiment suggested by the reviewer, we applied baclofen (90 s) onto the stratum radiatum to target GABA_B_R in astrocytes and we observed similar results as with interneuron depolarization. These data can be found in new Figure 3—figure supplement 1 and in the text: “To confirm GABA_B_Rs involvement in the astrocyte Ca^2+^ signals and the biphasic synaptic regulation we applied baclofen (90 s), a GABA_B_R agonist, into the stratum radiatum while monitoring astrocyte Ca^2+^ and synaptic currents in the pyramidal neuron (Figure 3—figure supplement 1). Baclofen application induced Ca^2+^ increases in astrocytes (n=25 astrocytes in n=4 slices; p=0.02; Figure 3—figure supplement 1) and a biphasic synaptic regulation (n=6; Pr potentiation p<0.001; Pr depression p=0.015; Figure 3—figure supplement 1), mimicking results obtained with interneuron depolarization.”

In addition, the following paragraph has been added in the Discussion section: “interestingly, the GABA_B_Rmediated biphasic synaptic regulatory phenomenon obtained with either interneuron depolarization or with direct baclofen application showed a slower time course than the responses obtained from neuronal ionotropic receptors. […] In fact, it is known that astrocyte Ca^2+^ signaling occurs in a slow and prolonged manner (Volterra, Liaudet and Savtchouk, 2014; Araque et al., 2014; Rusakov, 2015) probably due to a different expression of GABA_B_Rs by distinct astrocyte; to the fact that astrocyte responses require the build-up of extracellular GABA released from synaptic terminals, which may not reach all the monitored astrocytes; or the number of steps involved in activation of G protein-mediated signaling pathways and intracellular calcium mobilization”.

4) A key experiment suggesting the involvement of astrocytes involves analysis of synaptic potentiation and depression in IP_3_R2 ko mice. The authors should perform a control experiment to show that calcium uncaging in astrocytes in IP_3_R2 ko mice leads to the same biphasic change in Pr.

We thank the helpful suggestion of the reviewer. We have performed the suggested experiment and we found that both synaptic potentiation and depression can be rescued in IP_3_R2^-/-^ mice with Ca^2+^ uncaging. We have included the data in Figure 5 and in the text: “In addition, light-evoked Ca^2+^ uncaging rescued the biphasic synaptic modulation in the IP_3_R2^-/-^ mice (n=6 neurons; Pr potentiation p<0.003; Pr depression p<0.012; Figure 5), indicating that astrocytic glutamate and ATP/adenosine release are downstream the astrocyte Ca^2+^ signaling.”

5) Because GABA_B_ receptors are expressed by multiple cell types, the authors should consider performing these experiments in astrocyte specific conditional knockout mice. Such experiments would lend stronger support for the proposed model of neuromodulation.

We thank the reviewer for the valuable comment. As suggested, we have performed experiments in astroglial GABA_B_R^-/-^ mice that lacks GABA_B_Rs specifically in astrocytes. In these mice, we did not observe changes in astrocyte Ca^2+^ levels or synaptic transmission during or after interneuron stimulation. These new data support our hypothesis that GABA released by interneurons activates astrocytic GABA_B_Rs leading to a biphasic regulation of synaptic transmission. These data have been included in Figure 3 and in the text: “In addition, experiments performed in a transgenic mouse that lacks GABA_B_ receptors specifically in astrocytes (GABAB^-/-^) (Perea et al., 2016) also failed to induce Ca^2+^ elevations in astrocytes (n=53 astrocytes in n=10 slices; p=0.23; Figure 3) and the biphasic synaptic regulation (n=9 neurons; Pr potentiation p=0.75; Pr depression p=0.66; Figure 3). Taken together, these data indicates that GABAB-mediated astrocyte Ca^2+^ signaling is necessary to induce the interneuron-evoked potentiation and depression of synaptic transmission.”

Reviewer #2:[…] Showing that one astrocyte can have both effects on the same synapse is interesting, but it is hard to interpret the physiological meaning of the results the way they are currently presented. The incorporation of a model figure depicting the proposed circuit connections between the principal neurons, interneurons and astrocytes would help in interpretation, as would addressing the following questions about whether the neuronal stimulation parameters used correlate with endogenous firing properties of the neurons. These points do not need new experiments to address them.

We thank the reviewer for the positive comments and the helpful suggestions.

As suggested, a summary diagram outlining the major pathways involved has been included in Figure 2 and Figure 7.

In Figure 3 activation of the interneuron causes potentiation of an excitatory synapse (via the astrocyte). Is this known/expected to happen?

Interneuron-mediated potentiation of neurotransmitter release is not unexpected. It is known that astrocytes respond to GABA through the activation of GABA_B_Rs (Kang et al., 1998; Serrano et al., 2006; Mariotti et al., Glia 2016) that is also known to induce glutamate release form astrocytes (Kang et al., 1998; Perea et al., *eLife* 2016). We have recently described the underlying signaling mechanisms of GABA-induced synaptic potentiation and its functional consequences (Perea et al., 2016). This has been included in the text: “other studies have reported a GABA_B_R-mediated astrocytic glutamate release by direct interneuron stimulation that leads to a synaptic potentiation (Perea et al., 2015; Kang et al., 1998)”

Are the stimuli used for interneuron activation within the range of what is normally recorded in the hippocampus (20Hz for 90sec)?

This is an excellent question. Indeed, in vivo electrophysiological recordings have shown that hippocampal interneurons can fire action potentials at different frequencies that vary from 20 Hz during theta activity to >100 Hz during sharp wave ripples (see Pike et al., 2000 and Hu, Gan and Jonas, 2014). The following has been included in the text: “We monitored Ca^2+^ signals in *stratum radiatum* astrocytes and the SC-induced unitary EPSCs in the pyramidal cell, and stimulated interneurons with a train of short depolarizing pulses to trigger action potentials at 20 Hz (for 90 s), as previously reported for interneurons (Hu, Gan and Jonas, 2014; Pike et al., 2000) (Figure 3)”. We initially used this stimulation paradigm (20Hz for 90sec) to clearly reveal the effects, but the phenomena were further characterized using different frequencies and duration of the stimulus to test dependence of gliotransmitter release on these parameters (Figure 6).

Is there evidence that astrocytes undergo prolonged somatic calcium rises (seconds)? The effects on synapses on the whole seem to be on a slow timescale (minutes).

Indeed, somatic calcium signals in astrocytes display a relatively slow time scale (e.g., see the reviews discussing this issue, Araque et al., 2014, Volterra, Liaudet and Savtchouk, 2014, Rusakov, 2015), suggesting that astrocytes operate at a slower time scale than neurons. This has been discussed in the text: “Interestingly, the GABA_B_R-mediated biphasic synaptic regulatory phenomenon obtained with either interneuron depolarization or with direct baclofen application showed a slower time course than the responses obtained from neuronal ionotropic receptors. […] In fact, it is known that astrocyte Ca^2+^ signaling occurs in a slow and prolonged manner (Volterra, Liaudet and Savtchouk, 2014; Araque et al., 2014; Rusakov, 2015) probably due to a different expression of GABA_B_Rs by distinct astrocyte; to the fact that astrocyte responses require the build-up of extracellular GABA released from synaptic terminals, which may not reach all the monitored astrocytes; or the number of steps involved in activation of G protein-mediated signaling pathways and intracellular calcium mobilization.”

Does the size of the astrocyte calcium response correlate with the outcome at the synapse, i.e. just potentiation, or potentiation followed by depression? In Figure 3 the astrocyte calcium increases at the start of interneuron stimulation but has decayed back to baseline after 25sec even though the neuronal stimulus lasts for 90sec – is this a saturating effect?

This is an excellent question, but intrinsic experimental limitations associated with Ca^2+^ indicators prevent a clear answer. The amplitude of Ca^2+^ signal depends on the concentration of both intracellular Ca^2+^ and the fluorescent dye. Because the latter cannot be accurately estimated in brain slices, the recorded signal provides a qualitative estimation of the Ca^2+^ changes within the cell. We hence used the calcium event probability as a parameter indicative of astrocyte activity, but we refrained to establish correlations between changes in Pr and qualitative Ca^2+^ levels because they might be misleading.

Reviewer #3:The author (and others in the astrocyte field) has developed a consistent story over the past years describing a link between astrocyte calcium signals and alteration of neuronal synaptic responses. Based on pharmacological blockers the scenario is that CB1 receptor stimulation of astrocytes triggers calcium signals that in turn trigger glutamate release from astrocyte that enhances synaptic transmission. This excitatory response is transient and is mediated by mGluR1 receptors. The second pathway is proposed to be due to GABAB-mediated calcium transients in astrocytes (activated by GABA release from interneurons) causes ATP release from astrocytes and subsequent depression of neuronal activity due to adenosine formation from degradation of ATP. In the context of these previous studies that were reported in publications from this lab and others, the authors have designed experiments to determine whether one astrocyte can have similar actions of excitation and inhibition. They provide evidence supporting their conclusion that calcium signals in one identified astrocyte can lead to both types of synaptic modifications. However the authors have no data that ATP is released from astrocytes under these circumstances. Their data show that there is an adenosine-mediated depression of synaptic responses and the role for ATP release is inferred from other studies.

We thank the reviewer for the positive comments and useful suggestions that have helped strengthening our conclusions.

We agree with the reviewer that ATP release is indirectly assessed from the synaptic effects of its metabolic product adenosine. Astrocytes have been shown to be able to release adenosine and ATP in calcium-independent and calcium-dependent manner, respectively. Therefore, the gliotransmitter released is likely ATP because the synaptic depression it is absent in IP_3_R2^-/-^ mice, in which G protein mediated calcium signal is impaired in astrocytes, and astrocyte calcium uncaging is sufficient to induce the synaptic depression. Nevertheless, we have referred throughout the text to ATP/adenosine, because present study focuses on synaptic transmission regulation. We have clarified the issue in the Discussion: “Present results indicate that synaptic depression is mediated by activation of adenosine A_1_ receptors. […] Indeed, the synaptic depression is absent in IP_3_R2^-/-^ mice, in which G protein mediated Ca^2+^ signal is impaired in astrocytes (Figure 3), and astrocyte Ca^2+^ uncaging is sufficient to induce the synaptic depression (Figure 5).”.

The authors describe the results using several parameters such as potency, probability of release and synaptic efficacy. These are defined as "probability of release (ratio between the number of successes versus total number of stimuli); synaptic efficacy (mean peak amplitude of all responses) and synaptic potency (mean peak amplitude of the successes)". From the data presented in Figure 1 it appears that the impact on synaptic release is to change the failure rate for evoked epscs using this experimental paradigm. The authors describe the potency presumably as the amplitude of the evoked epscs excluding the failures. This part of the data presentation is confusing and should be clarified in the figure legend as well as the brief description in the Materials and methods. For example in Figure 2 the actual evoked responses are shown in the top traces and the average responses are shown in the lower traces. These averages must also include the failures as well as the evoked synaptic currents. For example the response in ND appear to have several responses with smaller amplitude and the responses during the HFS appear to be identical to the basal responses. This could be important because of the implication of the mechanism.

We apologize for the confusion. Indeed, the synaptic potency corresponds to EPSC amplitude of successful responses. A more detailed description of the synaptic parameters has been included: “Synaptic parameters analyzed were: probability of release (ratio between the number of successes versus total number of stimuli); synaptic efficacy (mean peak amplitude of all responses including successes and failures) and synaptic potency (mean peak amplitude of the successes, excluding failures) (Navarrete and Araque, 2010; Perea and Araque, 2007; Gomez-Gonzalo et al., 2014; Navarrete et al., 2012). A response was considered a failure if the amplitude of the current was < 3 times the standard deviation of the baseline current (0.7–2.3 pA) and was verified by visual inspection.”

The statistical analysis of synaptic transmission changes indicate that they are due to changes in the probability of neurotransmitter release without changes in the synaptic potency (Figure 2, Figure 3, Figure 4 and Figure 5). Therefore, traces in Figure 2 simply reflect the intrinsic variability of the EPSC amplitude (see Figure 1).

The average responses shown in the lower traces in Figure 1, Figure 2 and Figure 3 correspond with the synaptic efficacy, thus, they are the average of the upper traces including the failures and the successes. This will be stated in the figure legend in Figure 1: “Representative EPSC traces obtained with the minimal stimulation technique showing successes and failures in neurotransmitter release (n=20; upper panel) and the average trace acquired from those traces (synaptic efficacy; lower panel).”

The authors conclude that the release of transmitter at synapses is inhibited. However the impact may be the inhibition of the propagation of action potentials or failure of spikes to invade the presynaptic terminal instead of depression of synaptic release mechanisms.

We agree with the reviewer that synaptic depression may be due to failures in action potential invasion or in the release machinery. Regardless the potential presynaptic mechanism, which is out of the scope of the present work, the final synaptic outcome is the inhibition of transmitter release. We consequently refrained to propose any mechanism, and simply described the depression of the presynaptic transmitter release.

"These results indicate that interneuron-evoked stimulation of astrocytes leads to the potentiation and depression of synaptic transmission." This conclusion depends on the assumption that IP_3_R2 receptors are only in astrocytes and this assumption should be clearly stated. It is certainly true that IP_3_R2 are important for the calcium signals linked to Gq coupled GPCRs. The references cited here all deal with the loss of calcium signaling in astrocytes in the IP_3_R2 knockout mouse. However it is possible that there are IP_3_R2 receptors in some neurons or specific neuronal compartments. The conclusion that this is solely due to astrocytes would require an astrocyte selective knockout of IP_3_R2 and I am not aware of this evidence (although I could be wrong). The authors should also be aware of several studies showing that calcium signaling stills occurs in fine processes of astrocytes in the IP_3_R2^-/-^ transgenic mouse.

We agree with the reviewer. We are also aware of the reports showing calcium signal in astrocytic processes in these mice, although largely diminished. We therefore performed additional experiments to test the effects of blocking astrocyte calcium signal by loading the astrocyte network with GDPβS. In these conditions, astrocyte Ca^2+^ signal was abolished and the biphasic synaptic regulation was also absent. We have included these new results in Figure 3 and in the text: “We further tested astrocyte involvement on the synaptic biphasic regulation loading the astrocytic network with GDPβS through a patch pipette to specifically block G protein signaling in astrocytes. In GDPβS-loaded slices interneuron depolarization failed to induce astrocyte Ca^2+^ elevations (n=60 astrocytes in n=8 slices; p=0.53; Figure 3) and to induce the biphasic synaptic regulation (n=7 neurons; Pr potentiation p=0.3; Pr depression p=0.36; Figure 3).”

This conclusion is further supported by the demonstration that astrocyte Ca^2+^ uncaging is sufficient to induce the synaptic regulation in WT and IP_3_R2^-/-^ mice (Figure 5) and by the new results showing the lack of astrocyte Ca^2+^ elevations and synaptic regulation in the Astroglial GABA_B_R^-/-^ mice.

The authors have provided intriguing evidence that the pattern of action potential firing in interneurons will lead to the synaptic enhancement or synaptic depression. This is a remarkable finding that the authors have convincingly shown using a very impressive strategy involving paired recordings from identified cells in brain slices. The initial comparison in Figure 2 for neuronal depolarization (CB1 mediated enhancement) versus high frequency stimulation (GABAG mediated) seems puzzling. The authors could describe more clearly the rationale for these initial experiments then the progression to the interneuron specific responses. In particular why was no initial enhancement of synaptic responses observed in Figure 2, compared to the enhancement in Figure 3?

We thank the reviewer for the positive comment. The rationale of experiments shown in Figure 2 was to test whether single synapses can be regulated by two astrocyte-mediated mechanisms, i.e., endocannabinoid- and glutamate-mediated synaptic potentiation and GABA_B_- and adenosine-mediated synaptic depression. These experiments were designed to test whether a single synapse could be regulated by two different gliotransmitters (glutamate and ATP/adenosine). A more detailed clarification as well as schematic drawings (Figure 2) have been included: “Astrocytes are known to induce synaptic potentiation or heterosynaptic depression of CA3-CA1 synapses through the release of glutamate or ATP, respectively (Figure 2) (Navarrete and Araque, 2010; Perea and Araque, 2007; Pascual et al., 2005; Boddum et al., 2016; Zhang et al., 2003; Chen et al., 2013). To study whether a single synapse can be regulated by distinct gliotransmitters, we tested whether single synapses could undergo both synaptic regulatory phenomena (mediated by astroglial glutamate and ATP/adenosine).”; and: “While some studies proposed that heterosynaptic depression involves GABA_B_R-mediated astrocytic release of ATP after interneuron activation by HFS of SC (Figure 2) (Serrano et al., 2006), other studies have reported a GABA_B_Rmediated astrocytic glutamate release by direct interneuron stimulation that leads to a synaptic potentiation (Perea et al., 2016; Kang et al., 1998). Because those reports used two different forms of interneuron stimulation, i.e., direct interneuron depolarization or synaptic stimulation, we hypothesized that the differences in the type of gliotransmitter released (glutamate vs. ATP/adenosine) depend on the interneuron firing pattern.”

Regarding the question “In particular why was no initial enhancement of synaptic responses observed in Figure 2, compared to the enhancement in Figure 3?” it should be noted that data in Figure 2 and Figure 3 corresponds to two different stimulation paradigms. In Figure 2, astrocytes were activated by HFS of SC (as described by reference Serrano et al., 2006), which prevented the adequate simultaneously recording of synaptic transmission parameters during the HFS. We agree with the reviewer that an initial synaptic potentiation was expected during the HFS protocol, but it was concealed due to the experimental limitations of the protocol used. In contrast, such initial enhancement shown in Figure 3, could be monitored because astrocytes were activated by direct stimulation of whole-cell recorded interneurons, which did not interfere which the simultaneous analysis of EPSCs. For that reason, the rest of the experiments were performed using the strategy of paired recordings from pyramidal cells and interneurons.

[Editors' note: the author responses to the re-review follow.]

The manuscript has been revised to address the comments of reviewer 1 and 2 (reviewer 3 had no further comments).

Reviewer #1:The authors have conducted many additional experiments to address the concerns raised and have clarified some issues that were confusing. As a result, the manuscript is much improved. However, I have several additional comments.1) The bar graphs analyzing changes in release probability, potency, etc. show effects that were present during the initial time after the manipulation. Please indicate the frequency of test stimulation after the inducing stimulus (I might have missed it, but wasn't able to find this in the Materials and methods). Given that repetitive stimulation is required to assess these parameters, it is important to indicate how many responses were used to calculate these averages.

We apologize for the lack of sufficient details in the previous version of the manuscript. We have now clarified the issue. Synaptic parameters were acquired at 0.33Hz, as indicated in the Materials and methods section: “Paired pulses (2 ms duration with 50 ms interval) were continuously delivered at 0.33 Hz” and “during the interneuron depolarizing train, synaptic activity was continuously monitored at 0.33 Hz in the pyramidal neuron”.

The clarification of the number of EPSCs used to calculate the averages depicted in the bar graphs have now been included in the material and methods section as follows: “To determine synaptic changes, mean EPSCs (n = 60 EPSCs) recorded 3 min before the stimulus (basal) were compared with mean EPSCs (n = 20 EPSCs) after the stimulus (for potentiation) or after the initial transient synaptic potentiation subsided (for depression).”

2) The inclusion of astrocyte-specific deletion experiments for CB1Rs and GABABRs is a terrific addition to the study. However, the authors need to provide evidence that this genetic manipulation was effective by directly assessing the effects of a CB1 agonist and GABA/baclofen on Ca changes in astrocytes, similar to experiments shown in Figure 3—figure supplement 1. It should be noted that both manipulations (using GFAP-CreER) may lead to deletion of CB1 receptors from radial glia and thus eventually dentate granule neurons, with unknown consequences for the tri-synaptic hippocampal circuit.

We thank the reviewer for the positive comment and helpful suggestion. As indicated, we have tested the validity of both transgenic mouse lines. The corresponding results are reported in new Figure 2—figure supplement 2 and new Figure 3—figure supplement 2, and described in the text as follows:

“To confirm that astrocytes in GFAP-CB1-null mice selectively lacked CB1Rs, we tested the astrocyte responsiveness to the CB1R agonist WIN55-212,2 and to ATP in slices obtained from WT and GFAP-CB1-null mice. […] Both wildtype and GFAP-CB1-null mice showed similar DSI, indicating that CB1Rs were specifically absent in astrocytes in GFAPCB1-null mice (Figure 2—figure supplement 2).”

Likewise, “We first confirmed that GABA_B_-mediated astrocyte responses were absent in these transgenic mice. While astrocytes from GABA_B_-null mice did not respond to baclofen (n=74 astrocytes from n=6 slices, p=0.418), they responded to ATP (p<0.001), indicating that astrocytes were functional but selectively lacked GABA_B_R-mediated responses. Furthermore, neurons from wildtype and GABA_B_-null mice similarly responded to baclofen, indicating that GABA_B_Rs were specifically absent in astrocytes in GABA_B_-null mice (Figure 3—figure supplement 2)”.

Regarding the comment on the possible deletion of receptors in radial glia, we agree with the reviewer that manipulations using GFAP-CreER may lead to deletion of CB1 receptors from radial glia and eventually dentate granule neurons. Although this could lead to changes in circuit hippocampal function, we believe that our present results on the astrocyte regulation at the synapse level are not compromised by potential alterations of network function.

3) I am not convinced by either the data or the argumentation that the responses recorded represent exclusively single synapses. Although the author correctly indicates that this approach has been used previously, there remains considerable uncertainty that is inherent to the approach. Description of responses as single synapses should formally be reserved for paired recordings in which the number of functional connections can be directly established (typically through post-hoc immunolabeling). For example, the representative example data shown in Figure 1 highlights the difficulty in defining what is a single synapse response, as the response amplitude clearly varies continuously with no clear separation between what are deemed successes and failures (by coloration). The statements surrounding single synapses should be softened to allow for this uncertainty.

We agree with this reviewer (and also reviewer #2) about the inherent limitations of the minimal stimulation approach. Following the reviewer´s suggestion, we have softened the statements, indicating using the term “putative single synapses”. In addition, to clarify the issue, we have included the following paragraph in the Discussion: “Present results were obtained using the minimal stimulation approach, which is considered to allow monitoring the activity of single or very few synapses based on the relatively low amplitude and stability of the synaptic responses (Navarrete and Araque, 2010; Mariotti et al., 2018; Chen et al., 2013; Isaac et al., 1996; Raastad, 1995). However, the inherent limitation of the approach remained uncertain whether single or few synapses were activated. Nevertheless, the fact that the synaptic potency remained unchanged throughout the experiments indicates that the same synapse or synapses were monitored.”

4) I remain concerned about the use of data selection in the manuscript. Data should not be excluded simply because the cells did not respond as expected. Exclusion criteria should be clearly defined, for example, if the authors were to show that astrocytes varied in their expression of receptors, by challenging cells at the end of the experiment with a selective agonist. Regarding the source of variability, please indicate what proportion of astrocytes exhibited Ca increases in response to baclofen application.

We understand the reviewer´s concern about this sensitive issue. Our selection criteria was that synapses that did not show regulation in control conditions were not further considered. Considering that we were analyzing the regulation of single or few synapses, it is not unexpected that in some experiments the stimulus was inefficient to regulate some synapses, perhaps simply due to the absence of functional connectivity between the astrocyte stimulated and the synapse recorded. Because pulling together results from regulated or unregulated synapses would result in the concealment of the effects, as we have recently reported (Lines et al., 2017), the selection of the regulated synapses is therefore required. Nevertheless, the proportion of cells showing the phenomena are reported (Figure 2 and Figure 4—figure supplement 1).

To clarify the issue, the following paragraph has been included: “Figure 2 and Figure 4—figure supplement 1 shows the percentage of synapses undergoing potentiation, depression, both or no change. […] Hence, for pharmacology experiments and experiments using transgenic mice, a relatively large sample size is used, thus the absence of the phenomena can be assumed by never being observed in the recorded population.”

Regarding the proportion of astrocyte that responded to baclofen, we now indicate in the text the following sentence: “Baclofen application induced Ca^2+^ increases in 49 ± 8% of the recorded astrocytes (n=61 astrocytes in n=11 slices p<0.001; Figure 3—figure supplement 1).”

Reviewer #2:The authors have addressed the questions I raised in the first round of review in a satisfactory manner. The other reviewers raised more substantive concerns. In particular, the point reviewer 1 raised about the ability of minimal stimulation in the CA3-CA1 pathway to be monitoring the same synapse every time seems particularly important for the interpretation of the data – that one synapse is both potentiated and depressed by different gliotransmitters from the same astrocyte.

As indicated in the response to comment 3 of reviewer #1, we agree about the inherent limitations of the minimal stimulation approach. Following the reviewer´s suggestion, we have softened the statements, indicating using the term “putative single synapses”. In addition, to clarify the issue, we have included the following paragraph in the Discussion: “Present results were obtained using the minimal stimulation approach, which is considered to allow monitoring the activity of single or very few synapses based on the relatively low amplitude and stability of the synaptic responses (Navarrete and Araque, 2010; Mariotti et al., 2018; Chen et al., 2013; Isaac et al., 1996; Raastad, 1995). […] Nevertheless, the fact that the synaptic potency remained unchanged throughout the experiments indicates that the same synapse or synapses were monitored.”

Additionally, the new experiments using tamoxifen-inducible astrocyte-specific KO lines of different receptors don't appear to have any validation in this paper (there are references provided but one is from a different lab, and the other uses recombination in younger animals), so this is something that needs adding, or a more detailed explanation of validation of the models should be provided in the Materials and methods.

We thank the reviewer for the helpful suggestion. As indicated in the response to comment 2 of reviewer #1, we have experimentally tested the validity of both transgenic mouse lines. The corresponding results are reported in new Figure 2—figure supplement 2 and new Figure 3—figure supplement 2, and described in the text as follows:

“To confirm that astrocytes in GFAP-CB1-null mice selectively lacked CB1Rs, we tested the astrocyte responsiveness to the CB1R agonist WIN55-212,2 and to ATP in slices obtained from WT and GFAP-CB1-null mice. […] Both wildtype and GFAP-CB1-null mice showed similar DSI, indicating that CB1Rs were specifically absent in astrocytes in GFAPCB1-null mice (Figure 2—figure supplement 2).”

Likewise, “We first confirmed that GABA_B_-mediated astrocyte responses were absent in these transgenic mice. While astrocytes from GABA_B_-null mice did not respond to baclofen (n=74 astrocytes from n=6 slices, p=0.418), they responded to ATP (p<0.001), indicating that astrocytes were functional but selectively lacked GABA_B_R-mediated responses. Furthermore, neurons from wildtype and GABA_B_-null mice similarly responded to baclofen, indicating that GABA_B_Rs were specifically absent in astrocytes in GABA_B_-null mice (Figure 3—figure supplement 2)”